# How Discrete and Continuous Diffusion Meet: Comprehensive Analysis of Discrete Diffusion Models via a Stochastic Integral Framework

**Yinuo Ren**
ICME
Stanford University
`yinuoren@stanford.edu`

**Haoxuan Chen**
ICME
Stanford University
`haoxuanc@stanford.edu`

**Grant M. Rotskoff**
Department of Chemistry and ICME
Stanford University
`rotskoff@stanford.edu`

**Lexing Ying**
Department of Mathematics and ICME
Stanford University
`lexing@stanford.edu`

## Abstract

Discrete diffusion models have gained increasing attention for their ability to model complex distributions with tractable sampling and inference. However, the error analysis for discrete diffusion models remains less well-understood. In this work, we propose a comprehensive framework for the error analysis of discrete diffusion models based on Lévy-type stochastic integrals. By generalizing the Poisson random measure to that with a time-independent and state-dependent intensity, we rigorously establish a stochastic integral formulation of discrete diffusion models and provide the corresponding change of measure theorems that are intriguingly analogous to Itô integrals and Girsanov's theorem for their continuous counterparts. Our framework unifies and strengthens the current theoretical results on discrete diffusion models and obtains the first error bound for the $\tau$-leaping scheme in KL divergence. With error sources clearly identified, our analysis gives new insight into the mathematical properties of discrete diffusion models and offers guidance for the design of efficient and accurate algorithms for real-world discrete diffusion model applications.

## 1 Introduction

Diffusion and flow-based models designed for discrete distributions have gained significant attention in recent years due to their versatility and wide applicability across various domains. These models have been proposed and refined in several key works (Sohl-Dickstein et al., 2015; Austin et al., 2021; Floto et al., 2023; Hoogeboom et al., 2021a;b; Meng et al., 2022; Richemond et al., 2022; Sun et al., 2022; Santos et al., 2023). The appeal of such models stems from their potential to address challenging problems in fields like computational biology, where they have shown promise in tasks such as molecule, protein, and DNA sequence design (Seff et al., 2019; Alamdari et al., 2023; Avdeyev et al., 2023; Emami et al., 2023; Frey et al., 2023; Watson et al., 2023; Yang et al., 2023b; Campbell et al., 2024; Stark et al., 2024; Kerby & Moon, 2024; Yi et al., 2024). Additionally, these approaches have proven effective in combinatorial optimization (Li et al., 2024e), modeling retrosynthesis (Igashov et al., 2023), image synthesis (Lezama et al., 2022; Gu et al., 2022), text summarization (Dat et al., 2024) along with the generation of graph (Niu et al., 2020; Shi et al., 2020; Qin et al., 2023; Vignac et al., 2022), layout (Inoue et al., 2023; Zhang et al., 2023), motion (Chi et al., 2024; Lou et al., 2023), sound (Campbell et al., 2022; Yang et al., 2023a), image (Hu et al., 2022; Zhu et al., 2022; Ren et al., 2025a), speech (Wu et al., 2024) and text (He et al., 2022; Wu et al., 2023; Gong et al., 2023; Zheng et al., 2023; Zhou et al., 2023; Shi et al., 2024; Sahoo et al., 2024). Discrete diffusion models also synergize with other methodologies, including tensor networks (Causer et al., 2024), enhanced guidance mechanisms (Gruver et al., 2024; Nisonoff et al., 2024; Li et al., 2024d), structured preferential generation (Rissanen et al., 2024), and alternative metrics, *e.g.* the Fisher

information metric (Davis et al., 2024). These developments highlight the growing importance of discrete modeling in advancing both theoretical understanding and efficient implementations. For a more complete review of related literature, one may refer to the appendix of Ren et al. (2025a).

Partly due to the absence of a discrete equivalent to Girsanov's theorem, the error analysis for discrete diffusion models remains underdeveloped compared to their continuous counterparts. Existing theoretical work, including a Markov chain-based error analysis for $\tau$-leaping in total variation distance by Campbell et al. (2022) and further advancements for the particular state space $\mathbb{X} = \{0, 1\}^d$ by Chen & Ying (2024), are largely algorithm-specific and not quite easy to be generalized. In this work, our goal is to establish a comprehensive framework for analyzing discrete diffusion models through a stochastic analysis perspective, which is motivated by the theory of continuous diffusion models and different from previous works in many aspects. Drawing on tools from Lévy processes and methodologies for analyzing the simulations of chemical reactions (Li, 2007; Anderson et al., 2011), we extend Poisson random measures to those with evolving intensities, *i.e.* both time-inhomogeneous and state-dependent intensities (Protter, 1983), introduce Lévy-type stochastic integrals (Applebaum, 2009), and articulate corresponding change of measure theorems, which are analogous to the Itô integrals and Girsanov's theorem in continuous settings.

We further demonstrate that discrete diffusion models implemented via either the $\tau$-leaping or the uniformization scheme can be formulated as stochastic integrals w.r.t. Poisson random measures with evolving intensity, which leads to a unified framework for error analysis. This new stochastic integral-based framework, marking a first for discrete diffusion models, is especially convenient and straightforward for decomposing inference error into three parts: truncation, approximation, and discretization, drawing satisfying parallels with state-of-the-art theories for continuous diffusion model (Chen et al., 2022; 2023a; Benton et al., 2023a). Our approach thus provides intuitive explanations for the loss design and unifies the error analysis across both schemes, enhancing the comparative understanding of their convergence characteristics. Notably, we achieve stronger convergence results in KL divergence and relax some of the stringent assumptions previously required for the state space, the rate matrix, and the estimation of score functions, *etc.*, thereby paving the way for the analysis of a broader class of discrete diffusion models of interest, providing valuable tools for designing efficient and accurate algorithms tailored to the practical demands of discrete diffusion models in real-world applications, and facilitating the transfer of theoretical and practical insights between continuous and discrete diffusion models.

## 1.1 CONTRIBUTIONS

Our main contributions are summarized as follows:

- We develop a rigorous framework for discrete diffusion models using Lévy-type stochastic integrals based on the Poisson random measure with evolving intensity, which includes formulating discrete diffusion models into stochastic integrals and establishing change of measure theorems that facilitate explicit log-likelihood ratio calculations;
- Our framework extends to a comprehensive, continuous-time analysis for error decomposition in discrete diffusion models, drawing clear parallels with the methodologies used in continuous models and enabling more effective adaptations of techniques across different model types;
- We unify and fortify existing research on discrete diffusion models by deriving the first error bound for $\tau$-leaping in terms of KL divergence, stronger compared to earlier results in TV distance, and providing a comparative study of $\tau$-leaping and uniformization implementations.

## 1.2 RELATED WORKS

**Continuous Diffusion Models.** Continuous diffusion models have been one of the most active research areas in generative modeling. Earlier work on continuous diffusion models and probability flow-based models include (Sohl-Dickstein et al., 2015; Zhang et al., 2018; Song & Ermon, 2019; Ho et al., 2020; Song et al., 2020; 2021; Lipman et al., 2022; Liu et al., 2022; Albergo & Vanden-Eijnden, 2022; Albergo et al., 2023). It has shown state-of-the-art performance in various fields of science and engineering. For some recent work and comprehensive review articles, one may refer to (Xu et al., 2022; Yang et al., 2023c; Chan, 2024; Wang et al., 2023; Alakhdar et al., 2024; Chen et al., 2024b; Fan et al., 2024; Guo et al., 2024; Riesel et al., 2024; Zhu et al., 2024).

**Theory of Continuous Diffusion Models.** In addition to the huge success achieved by diffusion models in empirical studies, many works have also tried to establish sampling guarantees for diffu-

sion and probability flow-based models, such as (Tzen & Raginsky, 2019; Block et al., 2020; Benton et al., 2023b; Chen et al., 2023b; Mbacke & Rivasplata, 2023; Liang et al., 2024; Benton et al., 2024; Ren et al., 2025b). Regarding theoretical analysis of continuous diffusion models, (Lee et al., 2022) provided the first sampling guarantee under the smoothness and isoperimetry assumptions. Follow-up work removed such assumptions (Chen et al., 2022; 2023a; Lee et al., 2023) and obtained better convergence results (Benton et al., 2023a; Pedrotti et al., 2023; Li et al., 2023; 2024a;b). For the probability flow-based implementation, sampling guarantee was also established and further refined in many recent work (Chen et al., 2024c; Gao & Zhu, 2024; Huang et al., 2024; Li et al., 2024c)

## 2 PRELIMINARIES

In this section, we introduce the basic concepts of both continuous and discrete diffusion models and then roughly outline the error analysis for continuous diffusion models, which will serve as a reference for the error analysis for discrete diffusion models.

### 2.1 CONTINUOUS DIFFUSION MODELS

In diffusion models, the *forward process* is designed as an Itô process $(\boldsymbol{x}_t)_{0 \le t \le T}$ in $\mathbb{R}^d$ satisfying the following stochastic differential equation (SDE):

$$\mathrm{d}\boldsymbol{x}_t = \boldsymbol{b}_t(\boldsymbol{x}_t)\mathrm{d}t + \boldsymbol{g}_t\mathrm{d}\boldsymbol{w}_t, \text{ with } \boldsymbol{x}_0 \sim p_0, \tag{2.1}$$

where $(\boldsymbol{w}_t)_{t \ge 0}$ is a standard Brownian motion. The probability distribution of $\boldsymbol{x}_t$ is denoted by $p_t$, and the distribution $p_0$ at time $t = 0$ is the target distribution for sampling. The time-reversal $(\bar{\boldsymbol{x}}_s)_{0 \le s \le T}$ of (2.1) satisfies the *backward process*:

$$\mathrm{d}\bar{\boldsymbol{x}}_s = \left[ -\bar{\boldsymbol{b}}_s(\bar{\boldsymbol{x}}_s) + \bar{\boldsymbol{g}}_s\bar{\boldsymbol{g}}_s^\top \nabla \log \bar{p}_s(\bar{\boldsymbol{x}}_s) \right]\mathrm{d}s + \bar{\boldsymbol{g}}_s\mathrm{d}\boldsymbol{w}_s, \tag{2.2}$$

where $\bar{*}_s$ denotes $*_{T-s}$, with $\bar{p}_0 = p_T$ and $\bar{p}_T = p_0$.

One of the common choices for the drift $\boldsymbol{b}_t$ and the diffusion coefficient $\boldsymbol{g}$ is $\boldsymbol{b}_t(\boldsymbol{x}) = -\frac{1}{2}\beta_t\boldsymbol{x}_t$ and $\boldsymbol{g} = \sigma\sqrt{\beta_t}\boldsymbol{I}$, under which (2.1) is an Ornstein-Uhlenbeck (OU) process converging exponentially, *i.e.* $p_T \approx p_\infty := \mathcal{N}(0, \sigma^2\boldsymbol{I})$, and the forward process (2.1) and the backward process (2.2) reduce to the following form:

$$\mathrm{d}\boldsymbol{x}_t = -\frac{1}{2}\beta_t\boldsymbol{x}_t\mathrm{d}t + \sigma\sqrt{\beta_t}\mathrm{d}\boldsymbol{w}_t, \text{ and } \mathrm{d}\bar{\boldsymbol{x}}_s = \bar{\beta}_s \left[ \frac{1}{2}\bar{\boldsymbol{x}}_s + \sigma^2\nabla\log\bar{p}_s(\bar{\boldsymbol{x}}_s) \right]\mathrm{d}s + \sigma\sqrt{\bar{\beta}_s}\mathrm{d}\boldsymbol{w}_s. \tag{2.3}$$

In practice, the score function $\boldsymbol{s}_t(\boldsymbol{x}_t) := \nabla\log p_t(\boldsymbol{x}_t)$ is often estimated by a neural network $\hat{\boldsymbol{s}}_t^\theta(\boldsymbol{x}_t)$, where $\theta$ denotes the parameters, and trained via *denoising score-matching* (Hyvärinen & Dayan, 2005; Vincent, 2011):

$$\theta = \arg\min_\theta \int_0^T \psi_t \mathbb{E}_{\boldsymbol{x}_t \sim p_t} \left[ \left\| \nabla\log p_t(\boldsymbol{x}_t) - \hat{\boldsymbol{s}}_t^\theta(\boldsymbol{x}_t) \right\|^2 \right]\mathrm{d}t$$
$$= \arg\min_\theta \int_0^T \psi_t \mathbb{E}_{\boldsymbol{x}_0 \sim p_0} \left[ \mathbb{E}_{\boldsymbol{x}_t \sim p_{t|0}(\boldsymbol{x}_t|\boldsymbol{x}_0)} \left[ \left\| \nabla\log p_{t|0}(\boldsymbol{x}_t|\boldsymbol{x}_0) - \hat{\boldsymbol{s}}_t^\theta(\boldsymbol{x}_t) \right\|^2 \right] \right]\mathrm{d}t, \tag{2.4}$$

where $p_{t|0}(\boldsymbol{x}_t|\boldsymbol{x}_0)$ is the transition distribution from $\boldsymbol{x}_0$ to $\boldsymbol{x}_t$ under (2.3) with an explicit form as

$$\mathcal{N}(\boldsymbol{\mu}_t, \sigma_t^2\boldsymbol{I}), \text{ where } \boldsymbol{\mu}_t = \boldsymbol{x}_0 e^{-\frac{1}{2}\int_0^t \beta_t\mathrm{d}t} \text{ and } \sigma_t^2 = \sigma^2\left(1 - e^{-\int_0^t \beta_t\mathrm{d}t}\right), \tag{2.5}$$

and $\psi_t$ is a weighting function for the loss at time $t$. After obtaining the NN-based score function $\hat{\boldsymbol{s}}_t^\theta(\boldsymbol{x}_s)$, the backward process in (2.3) is approximated as:

$$\mathrm{d}\boldsymbol{y}_s = \left[ \frac{1}{2}\boldsymbol{y}_s + \hat{\boldsymbol{s}}_s^\theta(\boldsymbol{y}_s) \right]\mathrm{d}s + \mathrm{d}\boldsymbol{w}_s, \text{ with } \boldsymbol{y}_0 \sim q_0 = \mathcal{N}(0, \sigma^2\boldsymbol{I}). \tag{2.6}$$

### 2.2 DISCRETE DIFFUSION MODELS

In discrete diffusion models, one turns to consider a continuous-time Markov chain $(\boldsymbol{x}_t)_{0 \le t \le T}$ in a space $\mathbb{X}$ of finite cardinality as the *forward process*. We denote the probability distribution of $\boldsymbol{x}_t$ by a vector $\boldsymbol{p}_t \in \Delta^{|\mathbb{X}|}$, where $\Delta^{|\mathbb{X}|}$ denotes the probability simplex in $\mathbb{R}^{|\mathbb{X}|}$. Given the target distribution $\boldsymbol{p}_0$, the Markov chain satisfies the following master equation:

$$\frac{\mathrm{d}\boldsymbol{p}_t}{\mathrm{d}t} = \boldsymbol{Q}_t\boldsymbol{p}_t, \text{ where } \boldsymbol{Q}_t = (Q_t(y,x))_{x,y\in\mathbb{X}} \in \mathbb{R}^{|\mathbb{X}|\times|\mathbb{X}|} \tag{2.7}$$

is the rate matrix at time $t$. The rate matrix $\boldsymbol{Q}_t$ satisfies the following two conditions:

$$\text{(i) } Q_t(x,x) = -\sum_{y\neq x} Q_t(y,x), \ \forall x \in \mathbb{X}; \text{ (ii) } Q_t(x,y) \geq 0, \ \forall x \neq y \in \mathbb{X}.$$

In the following, we will use a shorthand notation $\widetilde{\boldsymbol{Q}}_t$ to denote the matrix $\boldsymbol{Q}_t$ with the diagonal elements set to zero. It can be shown that the corresponding backward process is of the same form but with a different rate matrix (Kelly, 2011):

$$\frac{\mathrm{d}\bar{\boldsymbol{p}}_s}{\mathrm{d}s} = \overline{\boldsymbol{Q}}_s\bar{\boldsymbol{p}}_s, \quad \text{where} \quad \overline{Q}_s(y,x) = \begin{cases} \frac{\bar{p}_s(y)}{\bar{p}_s(x)}\overleftarrow{Q}_s(x,y), & \forall x \neq y \in \mathbb{X}, \\ -\sum_{y'\neq x}\overline{Q}_s(y',x), & \forall x = y \in \mathbb{X}. \end{cases} \tag{2.8}$$

The rate matrix $\boldsymbol{Q}_t$ is often chosen to possess certain sparse structures such that the forward process converges to a simple distribution that is easy to sample from. Several popular choices include the uniform and absorbing transitions (Lou et al., 2024).

The common practice is to define the score function (or rather the score vector) as $\boldsymbol{s}_t(x) = (s_t(x,y))_{y\in\mathbb{X}} := \frac{\boldsymbol{p}_t}{p_t(x)}$, for any $x \in \mathbb{X}$ and estimate it by a neural network $\widehat{\boldsymbol{s}}_t^\theta(x)$, where the neural network $\theta$ is trained by minimizing the score entropy (Benton et al., 2022; Lou et al., 2024):

$$\begin{aligned} \theta &= \arg\min_\theta \int_0^T \psi_t \mathbb{E}_{x_t \sim p_t}\left[ \sum_{y\neq x}\left(-\log\frac{\widehat{s}_t^\theta(x,y)}{s_t(x,y)} - 1 + \frac{\widehat{s}_t^\theta(x,y)}{s_t(x,y)}\right) s_t(x,y)Q_t(x,y)\right]\mathrm{d}t \\ &= \arg\min_\theta \int_0^T \psi_t \mathbb{E}_{x_0 \sim p_0}\left[ \sum_{y\neq x}\left(-\frac{p_t(y|x_0)}{p_t(x|x_0)}\log\widehat{s}_t^\theta(x,y) + \widehat{s}_t^\theta(x,y)\right) Q_t(x,y)\right]\mathrm{d}t. \end{aligned} \tag{2.9}$$

Similar to the continuous case, the backward process is approximated by the continuous-time Markov chain with the following master equation with $\boldsymbol{q}_0 = \boldsymbol{p}_\infty$ and rate matrix $\widehat{\overline{\boldsymbol{Q}}}_s^\theta$:

$$\frac{\mathrm{d}\boldsymbol{q}_s}{\mathrm{d}s} = \widehat{\overline{\boldsymbol{Q}}}_s^\theta \boldsymbol{q}_s, \quad \text{where} \quad \widehat{\overline{Q}}_s^\theta(y,x) = \widehat{\overleftarrow{s}}_s^\theta(x,y)\overleftarrow{Q}_s(x,y), \ \forall x \neq y \in \mathbb{X}. \tag{2.10}$$

and sampling is accomplished by first sampling from the distribution $\boldsymbol{p}_\infty$ and then evolving the Markov chain accordingly.

## 2.3 Error Analysis of Continuous Diffusion Models

Before we proceed to the error analysis of discrete diffusion models, we first review that the error analysis of continuous diffusion models, which is often conducted by considering the following three error terms:

- **Truncation Error:** The error caused by approximating $p_T$ by $p_\infty$, which is often of the order $\mathcal{O}(d\exp(-T))$ due to exponential ergodicity;
- **Approximation Error:** The error caused by approximating the score function $\nabla \log p_t(\boldsymbol{x}_t)$ by a neural network $\widehat{\boldsymbol{s}}_t^\theta(\boldsymbol{x}_t)$, which is often assumed to be of order $\mathcal{O}(\epsilon)$, where $\epsilon$ is a small threshold, given a thorough training process;
- **Discretization Error:** The error caused by numerically solving the SDE (2.6) with Euler-Maruyama scheme or other schemes, *e.g.* exponential integrator (Zhang & Chen, 2022).

The total error is obtained from these three error terms with proper choices of the order of the time horizon $T$ and the design of the numerical scheme. We extract the following theorem from the state-to-the-art theoretical result (Benton et al., 2023a) for later comparison:

**Theorem 2.1** (Error Analysis of Continuous Diffusion Models). *Suppose the time discretization scheme $(s_i)_{i\in[0,N]}$ with $s_0 = 0$ and $s_N = T-\delta$ satisfies $s_{k+1} - s_k \leq \kappa(T - s_{k+1})$ for $k \in [0 : N-1]$. Assume $\mathrm{cov}(p_0) = \boldsymbol{I}$, and the score function $\nabla \log p_t(\boldsymbol{x}_t)$ is estimated by the neural network $\widehat{\boldsymbol{s}}_t^\theta(\boldsymbol{x}_t)$ with $\epsilon$-accuracy, i.e.*

$$\sum_{k=0}^{N-1}(s_{k+1} - s_k)\mathbb{E}_{\bar{\boldsymbol{x}}_{s_k} \sim \bar{p}_{s_k}}\left[\left\|\nabla \log \bar{p}_{s_k}(\bar{\boldsymbol{x}}_{s_k}) - \widehat{\overleftarrow{\boldsymbol{s}}}_{s_k}^\theta(\boldsymbol{x}_{s_k})\right\|^2\right] \leq \epsilon.$$

*Then under the following choice of the order of parameters*

$$T = \mathcal{O}(\log(d\epsilon^{-1})), \ \kappa = \mathcal{O}(d^{-1}\epsilon\log^{-1}(d\epsilon^{-1})), \ N = \mathcal{O}(d\epsilon^{-1}\log^2(d\epsilon^{-1})),$$

*we have $D_{\mathrm{KL}}(p_\delta \| \widehat{q}_{s_N}) \leq \epsilon$, where $\widehat{q}_{s_N}$ is the distribution of the approximate backward process (2.6) implemented with exponential integrator after $N$ steps.*

# 3 STOCHASTIC INTEGRAL FORMULATION OF DISCRETE DIFFUSION MODELS

In this section, we introduce the stochastic integral formulation of discrete diffusion models. The goal is to establish a path evolution equation analogous to Itô integral (or equivalently, stochastic differential equations) with the master equation (2.7) and (2.8) analogous to the Fokker-Planck equation in the continuous case.

## 3.1 POISSON RANDOM MEASURE WITH EVOLVING INTENSITY

In the following, the Poisson distribution with expectation $\lambda$ is denoted by $\mathcal{P}(\lambda)$.

**Definition 3.1** (Poisson Random Measure with Evolving Intensity). *Let $(\Omega, \mathcal{F}, \mathbb{P})$ be a probability space and $(\mathbb{X}, \mathcal{B}, \nu)$ be a measure space and $\lambda_t(y)$ is a non-negative predictable process on $\mathbb{R}^+ \times \mathbb{X} \times \Omega$ satisfying for any $T > 0$, $\int_0^T \int_{\mathbb{X}} 1 \vee |y| \vee |y|^2 \lambda_t(y) \nu(\mathrm{d}y) \mathrm{d}t < \infty$, a.s.. The random measure $N[\lambda](\mathrm{d}t, \mathrm{d}y)$ on $\mathbb{R}^+ \times \mathbb{X}$ is called a Poisson random measure with evolving intensity $\lambda_t(y)$ if*

*(i) For any $B \in \mathcal{B}$ and $0 \leq s < t$, $N[\lambda]((s, t] \times B) \sim \mathcal{P}\left(\int_s^t \int_B \lambda_\tau(y) \nu(\mathrm{d}y) \mathrm{d}\tau\right)$;*

*(ii) For any $t \geq 0$ and disjoint sets $\{B_i\}_{i \in [n]} \subset \mathcal{B}$, $\{N_t[\lambda](B_i) := N[\lambda]((0, t] \times B_i)\}_{i \in [n]}$ are independent stochastic processes.*

Well-definedness of the Poisson random measure with evolving intensity is non-trivial and we refer readers to Appendix A for more details and discussions. Intuitively, Poisson random measures randomly assign jumps to intervals along the evolution, with the number of points in each region following a Poisson distribution, while those with evolving intensity not only assign jumps but also locations controlled by the intensity function $\lambda_t(y)$. In the following, we will denote the filtration generated by the Poisson random measure $N[\lambda](\mathrm{d}t, \mathrm{d}y)$ by $(\mathcal{F}_t)_{t \geq 0}$.

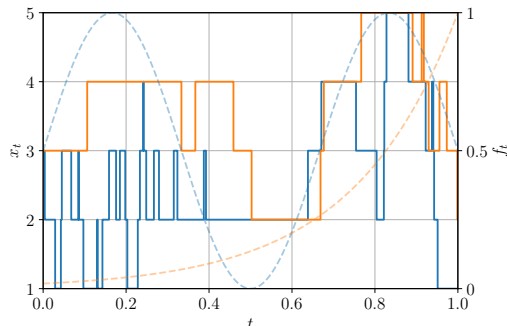

Figure 1: Example trajectories of stochastic integrals (3.1) w.r.t. Poisson random measure with different evolving intensities. The intensity is chosen as $\lambda_t(y) = 50 f_t$ if $|y - x_{t-}| = 1$ or otherwise 0, as shown in dashed lines. Intuitively, $\lambda_t$ controls the rate of jumps at time $t$ and location $y$.

The Poisson random measure defined above admits similar properties as the standard Poisson random measure and the Brownian motion. In particular, one can extend the Itô integral to the Lévy-type stochastic integral w.r.t. Poisson random measure with evolving intensity for predictable processes. It also admits Itô isometry, Itô's formula (Theorem A.10), and Lévy's characterization theorem (Theorem A.9), for which we refer readers to Appendix A.2 for details.

Now we turn to the setting of discrete diffusion models, where the state space $\mathbb{X}$ is finite endowed with the natural $\sigma$-algebra $\mathcal{B} = 2^{\mathbb{X}}$ and the counting measure $\nu = \sum_{y \in \mathbb{X}} \delta_y$.

**Proposition 3.2** (Stochastic Integral Formulation of Discrete Diffusion Models). *The forward process in discrete diffusion models (2.7) can be represented by the following stochastic integral:*

$$x_t = x_0 + \int_0^t \int_{\mathbb{X}} (y - x_{\tau-}) N[\lambda](\mathrm{d}\tau, \mathrm{d}y), \text{ with } \lambda_\tau(y) = \widetilde{Q}_\tau(y, x_{\tau-}), \tag{3.1}$$

*and the backward process in discrete diffusion models (2.8) can be represented by the following stochastic integral:*

$$\bar{x}_t = \bar{x}_0 + \int_0^t \int_{\mathbb{X}} (y - \bar{x}_{\tau-}) N[\mu](\mathrm{d}\tau, \mathrm{d}y), \text{ with } \mu_\tau(y) = \bar{s}_\tau(\bar{x}_{\tau-}, y) \widetilde{Q}_\tau(\bar{x}_{\tau-}, y), \tag{3.2}$$

*where $X_{t-}$ denotes the left limit of a càdlàg process $X_t$ at time $t$.*

The proof of Proposition 3.2 is provided in Appendix A.4. Figure 1 illustrates the stochastic integral in (3.1), comparing two different intensity functions. It is evident that jumps occur more frequently

in regions where the intensity is higher and less frequently where it is lower. Additionally, the intensity function, which encapsulates key information about the stochastic process, is often much easier and more straightforward to analyze compared to the complex trajectories of the process.

We would like to remark that the stochastic integral formulation in Proposition 3.2 is tantalizingly close to the Itô integral in continuous diffusion models in the form of stochastic differential equations (*cf.* (2.1) and (2.2)). Recalling that in the continuous case, Girsanov's theorem is applied to SDEs for deriving the score-matching loss (2.4) and also for the error analysis by associating the loss with the KL divergence, one may wonder if similar techniques can be applied to discrete diffusion models. The following section gives an affirmative answer to this question by providing a change of measure for Poisson random measures with evolving intensity, which is the theoretical foundation of our stochastic integral-based error analysis framework for discrete diffusion models.

## 3.2 Change of Measure

The following theorem provides a change-of-measure argument for stochastic integrals w.r.t. Poisson random measures with evolving intensity, analogous to Girsanov's theorem for Itô integrals w.r.t. Brownian motions.

**Theorem 3.3** (Change of Measure for Poisson Random Measure with Evolving Density). *Let $N[\lambda](\mathrm{d}t, \mathrm{d}y)$ be a Poisson random measure with evolving intensity $\lambda_t(y)$ in the probability space $(\Omega, \mathcal{F}, \mathbb{P})$, and $h_t(y)$ be a positive predictable process on $\mathbb{R}^+ \times \mathbb{X} \times \Omega$. Suppose the following exponential process is a local $\mathcal{F}_t$-martingale:*

$$Z_t[h] := \exp\left( \int_0^t \int_\mathbb{X} \log h_\tau(y) N[\lambda](\mathrm{d}\tau \times \mathrm{d}y) - \int_0^t \int_\mathbb{X} (h_\tau(y) - 1)\lambda_\tau(y)\nu(\mathrm{d}y)\mathrm{d}\tau \right), \quad (3.3)$$

*and $\mathbb{Q}$ is another probability measure on $(\Omega, \mathcal{F})$ such that $\mathbb{Q} \ll \mathbb{P}$ with Radon-Nikodym derivative $\mathrm{d}\mathbb{Q}/\mathrm{d}\mathbb{P}|_{\mathcal{F}_t} = Z_t[h]$. Then the Poisson random measure $N[\lambda](\mathrm{d}t, \mathrm{d}y)$ under the measure $\mathbb{Q}$ is a Poisson random measure with evolving intensity $\lambda_t(y)h_t(y)$.*

Intuitively, Theorem 3.3 depicts the relations between the Random-Nikodym derivative ($Z_t[h]$) of the probability measures on which two Poisson random measures with different intensities are defined. $Z_t[h]$ also roughly translates to the likelihood ratio between two paths generated with different intensities, with which one could work on stronger convergence results, *e.g.* KL divergence.

It is straightforward to derive the following corollary, which was derived in (Benton et al., 2022) with a different technique with Feller processes and adopted in (Lou et al., 2024; Chen & Ying, 2024) in the design of loss functions for the neural network training:

**Corollary 3.4** (Equivalence between KL Divergence and Score Entropy-based Loss Function). *Let $\breve{p}_{0:T}$ and $q_{0:T}$ be the path measures of the backward process (2.8) and the approximate backward process (2.10), then it holds that*

$$D_{\mathrm{KL}}(\breve{p}_T \| q_T) \le D_{\mathrm{KL}}(\breve{p}_{0:T} \| q_{0:T})$$

$$= D_{\mathrm{KL}}(\breve{p}_0 \| q_0) + \mathbb{E}\left[ \int_0^T \int_\mathbb{X} K\left( \frac{\breve{s}_\tau^\theta(\breve{x}_{\tau-}, y)}{\breve{s}_\tau(\breve{x}_{\tau-}, y)} \right) \breve{s}_\tau(\breve{x}_{\tau-}, y) \widetilde{Q}_\tau(\breve{x}_{\tau-}, y)\nu(\mathrm{d}y)\mathrm{d}\tau \right], \quad (3.4)$$

*where $K(x) = x - 1 - \log x \ge 0$, and the expectation is taken w.r.t. paths generated by the backward process (3.2). Consequently, minimizing the loss function (2.9) for discrete diffusion models is equivalent to minimizing the KL divergence between the path measures of the ground truth and the approximate backward process.*

Proofs of the change of measure-related arguments above will be provided in Appendix A.3. One should recall that in the continuous case with Itô integrals, the proximity of two paths in KL divergence only requires a small difference between the drift terms by Girsanov's theorem, and thus the score function can be trained with the mean squared error loss (2.4) (Song et al., 2020), while in the discrete case, the proximity of two paths requires the likelihood ratio to be close to one, accounting for a more complicated score entropy design (2.9) (Benton et al., 2022; Lou et al., 2024).

## 4 Error Analysis of Discrete Diffusion Models

In this section, we firstly review two different implementations of the discrete diffusion models, namely $\tau$-leaping (Gillespie, 2001) and uniformization (Van Dijk, 1992), derive their stochastic integral formulations as in Proposition 3.2, and provide our main results for their error analysis.

## 4.1 ALGORITHMS

### 4.1.1 $\tau$-LEAPING.

A straightforward algorithm for simulating the backward process is to discretize the integral in (3.2) with an Euler-Maruyama scheme. This leads to the $\tau$-leaping algorithm summarized in Algorithm 1. The main idea is to employ a predetermined time discretization scheme and approximate the inference process within each interval using the intensity observed at the time and location corresponding to the start of the interval.

---

**Algorithm 1:** $\tau$-Leaping Algorithm for Discrete Diffusion Model Inference

**Input:** $\widehat{y}_0 \sim q_0$, time discretization scheme $(s_i)_{i \in [0:N]}$ with $s_0 = 0$ and $s_N = T - \delta$, intensity function $\widehat{\mu}_s^\theta$ defined in Proposition 4.1, and neural network-based score function estimation $\widehat{s}_t^\theta$.

**Output:** A sample $\widehat{y}_{s_N} \sim \widehat{q}_{t_N}$.

1 **for** $n = 0$ **to** $N - 1$ **do**

2 $\quad \widehat{y}_{s_{n+1}} \leftarrow \widehat{y}_{s_n} + \sum_{y \in \mathbb{X}} (y - \widehat{y}_{s_n}) \mathcal{P}(\widehat{\mu}_{s_n}^\theta(y)(s_{n+1} - s_n));$  (4.1)

3 **end**

---

As shown in the following proposition, $\tau$-leaping can be formulated as a stochastic integral.

**Proposition 4.1** (Stochastic Integral Formulation of $\tau$-Leaping). *The $\tau$-leaping algorithm (Algorithm 1) is equivalent to solving the following stochastic integral equation:*

$$\widehat{y}_s = \widehat{y}_0 + \int_0^s \int_{\mathbb{X}} (y - \widehat{y}_{\lfloor \tau \rfloor -}) N[\widehat{\mu}_{\lfloor \cdot \rfloor}^\theta](\mathrm{d}\tau, \mathrm{d}y),$$  (4.2)

*where the evolving intensity $\widehat{\mu}_\tau^\theta(y)$ is given by $\widehat{\mu}_{\lfloor \tau \rfloor}^\theta(y) = \overleftarrow{\widehat{s}}_{\lfloor \tau \rfloor}^\theta(\widehat{y}_{\lfloor \tau \rfloor -}, y) \widetilde{Q}_{\lfloor \tau \rfloor}(\widehat{y}_{\lfloor \tau \rfloor -}, y) = \widehat{\mu}_{s_n}^\theta(y),$ in which we used the symbol $\lfloor \tau \rfloor = s_n$ for $\tau \in [s_n, s_{n+1})$. We will call the process $\widehat{y}_s$ the* interpolating process *of the $\tau$-leaping algorithm and denote the distribution of $\widehat{y}_s$ by $\widehat{q}_s$.*

In general, the stochastic integral associated with $\tau$-leaping (4.2) is computed using a piecewise constant intensity rather than the original continuous intensity. This approximation introduces discretization error as a trade-off for more efficient implementation. We refer to Remark A.13 for further justification of the $\tau$-leaping algorithm.

### 4.1.2 UNIFORMIZATION

Another algorithm considered for simulating the backward process in discrete diffusion models is uniformization. The algorithm is summarized in Algorithm 2, in which $\sigma_{(m)}$ denotes the $m$-th order statistic of the $M$ uniform random variables on $[0, 1]$, and the randomness in (4.3) should be understood as sampling a categorical distribution and updating the state accordingly.

---

**Algorithm 2:** Uniformization Algorithm for Discrete Diffusion Model Inference

**Input:** $\widehat{y}_0 \sim q_0$, time discretization scheme $(s_b)_{b \in [0,N]}$ with $s_0 = 0$ and $s_B = T - \delta$, intensity upper bound process $\overline{\lambda}_s$, intensity function $\widehat{\mu}_s^\theta$ defined in Proposition 4.1, and neural network-based score function estimation $\widehat{s}_t^\theta$.

**Output:** A sample $x_{s_B} \sim q_{t_B}$.

1 **for** $b = 0$ **to** $B - 1$ **do**

2 $\quad M \sim \mathcal{P}(\overline{\lambda}_{s_{b+1}}(s_{b+1} - s_b)), \sigma_m \sim \mathrm{Unif}([0, 1])$ for $m \in [M];$

3 $\quad$ **for** $m = 1$ **to** $M$ **do**

4 $\qquad \widehat{y}_{s_b + \sigma_{(m)}} \leftarrow \begin{cases} y, & \text{with prob. } \widehat{\mu}_{s_b + \sigma_{(m)}}^\theta(y)/\overline{\lambda}_{s_{b+1}}, \text{ for } y \in \mathbb{X}, \\ \widehat{y}_{s_b}, & \text{with prob. } 1 - \sum_{y \in \mathbb{X}} \widehat{\mu}_{s_b + \sigma_{(m)}}^\theta(y)/\overline{\lambda}_{s_{b+1}}; \end{cases}$  (4.3)

5 $\quad$ **end**

6 **end**

---

The main idea is to simulate the backward process by a Poisson random measure with a piecewise constant intensity upper bound process and then sample the behavior of each jump according to the

intensity $\widehat{\mu}_s^\theta(y)$ at time $s$. The uniformization algorithm also admits a stochastic integral formulation, as shown in the following proposition.

**Proposition 4.2** (Stochastic Integral Formulation of Uniformization). *Under the block discretization scheme $(s_b)_{b \in [0,B]}$ with $s_0 = 0$ and $s_B = T - \delta$, and for any $s \in (s_b, s_{b+1}]$, we define the piecewise constant intensity upper bound process by $\overline{\lambda}_s = \sup_{s \in (s_b, s_{b+1}]} \int_{\mathbb{X}} \widehat{\mu}_s^\theta(y)\nu(\mathrm{d}y)$. Then the uniformization algorithm (Algorithm 2) is equivalent to solving the following stochastic integral equation in the augmented measure space $(\mathbb{X} \times [0, \overline{\lambda}], \mathcal{B} \otimes \mathcal{B}([0, \overline{\lambda}]), \nu \otimes m)$:*

$$y_s = y_0 + \int_0^s \int_{\mathbb{X}} \int_{\mathbb{R}} (y - y_{\tau^-}) \mathbf{1}_{0 \leq \xi \leq \int_{\mathbb{X}} \widehat{\mu}_\tau^\theta(y)\nu(\mathrm{d}y)} N[\widehat{\mu}^\theta](\mathrm{d}\tau, \mathrm{d}y, \mathrm{d}\xi), \tag{4.4}$$

*where the evolving intensity $\widehat{\mu}_\tau^\theta(y)$ is given by $\widehat{\mu}_\tau^\theta(y) = \overleftrightarrow{\widehat{s}}_\tau^\theta(\widehat{y}_{\tau^-}, y)\widetilde{Q}_\tau(\widehat{y}_{\tau^-}, y)$.*

Based on Proposition 4.2, one can show that the uniformization algorithm simulates the backward process in discrete diffusion models accurately (*cf.* Theorem A.15), and the proofs of the claims above will be provided in Appendix A.4.

## 4.2 Assumptions

We need the following assumptions to ensure the well-definedness of discrete diffusion models. For simplicity, we assume the rate matrix $\boldsymbol{Q}_t$ is time-homogeneous and symmetric, *i.e.* $\boldsymbol{Q}_t = \boldsymbol{Q}$ for any $t \geq 0$. In fact, the results can be easily extended to the time-inhomogeneous case of the family $\boldsymbol{Q}_t = \beta_t \boldsymbol{Q}$ with a rescaling factor $\beta_t$, and asymmetric cases will be left for future works.

**Assumption 4.3** (Regularity of the Rate Matrix). *The rate matrix $\boldsymbol{Q}$ satisfies:*

 (i) *For any $x, y \in \mathbb{X}$, $Q(x, y) \leq C$ and $\underline{D} \leq -Q(x, x) \leq \overline{D}$ for some constants $C, \underline{D}, \overline{D} > 0$;*

 (ii) *The modified log-Sobolev constant $\rho(\boldsymbol{Q})$ of the rate matrix $\boldsymbol{Q}$ (cf. Definition B.6) is lower bounded by $\rho > 0$.*

Statement (i) assumes the regularity of the rate matrix, which is often trivially satisfied in many applications, while Statement (ii) ensures the exponential convergence of the forward process in discrete diffusion models. In general, $\rho(\boldsymbol{Q})$ may depend on the connectivity and other structures of the corresponding graph $\mathcal{G}(\boldsymbol{Q})$ (*cf.* Definition B.1). Such lower bound has been obtained for specific graphs (*e.g.* Example B.10 and B.11), and general results are in active research (Saloff-Coste, 1997; Bobkov & Tetali, 2006). We refer readers to Appendix B for further discussions on the literature of the modified log-Sobolev constant, as well as its relation to the spectral gap, the mixing time, *etc.*.

**Assumption 4.4** (Bounded Score). *The true score function satisfies $s_t(x, y) \lesssim 1 \vee t^{-1}$, while the learned score function satisfies $\widehat{s}_s^\theta(x, y) \in (0, M]$, for any $x, y \in \mathbb{X}$.*

The first part on the asymptotic behavior of the true score corresponds to the estimation $\mathbb{E}[\|\boldsymbol{s}_t\|^2] \sim \mathbb{E}[\|\boldsymbol{x}_t - \boldsymbol{\mu}_t\|^2/\sigma_t^2] \sim 1 \vee t^{-1}$ in the continuous case (Chen et al., 2023a, Assumption 1) and further justification is provided in Remark B.3. The bound on the estimated score can be easily satisfied by adding truncation in post-processing in the implementation of the NN-based score estimator.

**Assumption 4.5** (Continuity of Score Function). *For any $t > 0$ and $y \in \mathbb{X}$ such that $Q(x_{t^-}, y) > 0$, we have $\left|\frac{\mu_{t^+}(y)}{\mu_t(y)}\right| := \left|\frac{p_t(x_{t^-})Q(x_t, y)}{p_t(x_t)Q(x_{t^-}, y)} - 1\right| \lesssim 1 \vee t^{-\gamma}$, for some exponent $\gamma \in [0, 1]$.*

Assumption 4.5 corresponds to the Lipschitz continuity of the score function (*cf.* (Chen et al., 2022, Assumption 1), (Chen et al., 2023a, Assumption 3)) for continuous diffusion models, and is in light of the postulation that adjacent vertices should have close score function and intensity values. In the worse case, assume $Q(x, y) = \Theta(1)$, then a naïve bound would be $\left|\frac{\mu_{t^+}(y)}{\mu_t(y)}\right| \lesssim |s_t(x_t, x_{t^-})| \lesssim 1 \vee t^{-1}$ with $\gamma = 1$. However, when the initial distribution is both upper and lower bounded, $\gamma$ may be as small as 0, and we plan to investigate how this (local) continuity of the score function affects the overall performance of discrete diffusion models.

**Assumption 4.6** ($\epsilon$-accurate Score Estimation). *The score function $s_t(x_t)$ is estimated by the neural network $\widehat{s}_t^\theta(x_t)$ with $\epsilon$-accuracy, i.e.*

$$\sum_{n=0}^{N-1} (s_{n+1} - s_n)\mathbb{E}\left[\int_{\mathbb{X}} K\left(\frac{\overleftrightarrow{\widehat{s}}_{s_n}^\theta(\overline{x}_{s_n^-}, y)}{\overleftrightarrow{s}_{s_n}(\overline{x}_{s_n^-}, y)}\right) \overleftrightarrow{s}_{s_n}(\overline{x}_{s_n^-}, y)\widetilde{Q}(\overline{x}_{s_n^-}, y)\nu(\mathrm{d}y)\right] \leq \epsilon.$$

This assumption assumes the expressive power and sufficient training of the NN-based score estimator and is standard in diffusion model-related theories (Chen et al., 2022; 2023a; Benton et al., 2023a; Chen et al., 2024c).

## 4.3 ERROR ANALYSIS

Our main results are presented below for each algorithm introduced in Section 4.1.

### 4.3.1 $\tau$-LEAPING

**Theorem 4.7** (Error Analysis of $\tau$-Leaping). *Suppose the time discretization scheme $(s_i)_{i \in [0,N]}$ with $s_0 = 0$ and $s_N = T - \delta$ satisfies for $k \in [0 : N - 1]$, $s_{k+1} - s_k \leq \kappa \left( 1 \vee (T - s_{k+1})^{1+\gamma-\eta} \right)$, where the exponent $\eta$ satisfies $\gamma < \eta \lesssim 1 - T^{-1}$ when $\gamma < 1$, and $\eta = 1$ when $\gamma = 1$. With Assumptions 4.3, 4.4, 4.5, and 4.6 in place, we have the following error bound with probability $1 - \mathcal{O}(\epsilon)$*

$$D_{\mathrm{KL}}(p_\delta \| \widehat{q}_{T-\delta}) \lesssim \exp(-\rho T) \log |\mathbb{X}| + \epsilon + \overline{D}^2 \kappa T,$$

*and under the following choice of the order of parameters:*

$$T = \mathcal{O}\left( \frac{\log\left(\epsilon^{-1} \log |\mathbb{X}|\right)}{\rho} \right), \ \kappa = \mathcal{O}\left( \frac{\epsilon \rho}{\overline{D}^2 \log(\epsilon^{-1} \log |\mathbb{X}|)} \right), \ \delta = \begin{cases} 0, & \gamma < 1, \\ \Omega(e^{-\sqrt{T}}), & \gamma = 1, \end{cases} \quad (4.5)$$

*we have $D_{\mathrm{KL}}(p_\delta \| \widehat{q}_{T-\delta}) \lesssim \epsilon$ with $N = \kappa^{-1}T = \mathcal{O}\left( \frac{\overline{D}^2 \rho^2 \log^2\left(\epsilon^{-1} \log |\mathbb{X}|\right)}{\epsilon} \right)$ total steps.*

The conclusions and detailed proof (as provided in Appendix C), whose sketch is given in Appendix C.1, are analogous to the error bound for continuous diffusion models (*cf.* Theorem 2.1), as a summation of the truncation, approximation, and discretization errors as outlined in Section 2.3. We would like to point out the main differences between continuous and discrete diffusion models:

- **Truncation Error**: While the Ornstein-Uhlenbeck process converges exponentially fast in the continuous diffusion models, the exponential convergence of the forward process in discrete diffusion models is non-trivial for general graphs $\mathcal{G}(\boldsymbol{Q})$, for which the lower bound on the modified log-Sobolev constant $\rho$ is one of the sufficient conditions. In practice, the exponential convergence of the forward process should be verified for the specific problem at hand;
- **Discretization Error**: In continuous diffusion models, the analysis of the discretization error is based on the Itô integral and Girsanov's theorem, while for discrete diffusion models, the stochastic integral framework, including the Poisson random measure with evolving intensity (*cf.* Definition 3.1) and change of measure (*cf.* Theorem 3.3) that we developed above, is employed instead.

**Remark 4.8** (Remark on Early Stopping). *As in Assumption 4.4, the true score function may exhibit singular behavior as $s \to T$, due to possible vacancy in the target distribution $\boldsymbol{p}_0$. To handle this singularity, two different regimes are considered for the time discretization scheme depending on the continuity parameter $\gamma$ of the score function (Assumption 4.5). The main intuition is that (a) in the worse case $\gamma = 1$, early stopping at time $s = T - \delta$ is necessary; (b) if the target distribution $\boldsymbol{p}_0$ is such well-posed (e.g. both upper and lower bounded) and the rate matrix $\boldsymbol{Q}$ is constructed in a way that the score exhibits certain (local) continuity reflected by $\gamma < 1$, one may choose an appropriate shrinkage $\eta$, with which finite discretization error can be achieved with finite steps.*

In Theorem 4.7, the coefficient $\overline{D}$ roughly translates to the dimension $d$ when the discrete diffusion model is applied to $\mathbb{X} = [S]^d$, where $S$ is the number of states along each dimension. For example, when each state is connected to those at a Manhattan distance of 1 with unit weight, we have $C = 1$, $\overline{D} = 2d$, $\log |\mathbb{X}| = d \log S$ hold in Assumption 4.3. Plugging $\overline{D} = \log |\mathbb{X}| = \mathcal{O}(d)$ into the results, we obtain that the total number of steps $N = \widetilde{\mathcal{O}}(d^2)$, where $\widetilde{\mathcal{O}}$ denotes the order up to logarithmic factors. This recovers the dependency described in (Campbell et al., 2022, Theorem 1) for $\tau$-leaping with a completely different set of techniques, and importantly, our results do not rely on strong assumptions such as a uniform bound on the true score. We also reduce assumption stringency by relating our assumption on the estimation error (Assumption 4.6) more closely to the practical training loss rather than requiring an $L^\infty$-accurate score estimation error. Most notably, we provide the first convergence guarantees for $\tau$-leaping in KL divergence, which is stronger than the total variation distance, for discrete diffusion models.

### 4.3.2 UNIFORMIZATION

The error analysis of the uniformization algorithm requires the following modified assumption on the accuracy of the learned score function $\widehat{s}_t^\theta(x_t)$:

**Assumption 4.6'** ($\epsilon$-accurate Learned Score). *The score function $s_t(x_t)$ is estimated by the neural network $\widehat{s}_t^\theta(x_t)$ with $\epsilon$-accuracy, i.e.*

$$\mathbb{E}\left[\int_0^{T-\delta}\int_{\mathbb{X}} K\left(\frac{\overleftarrow{\widehat{s}}_s^\theta(\overleftarrow{x}_{s^-},y)}{\overleftarrow{s}_s(\overleftarrow{x}_{s^-},y)}\right)\overleftarrow{s}_s(\overleftarrow{x}_{s^-},y)\widetilde{Q}(\overleftarrow{x}_{s^-},y)\nu(\mathrm{d}y)\right]\mathrm{d}s \le \epsilon.$$

**Theorem 4.9** (Error Analysis of Uniformization). *Suppose the block discretization scheme $(s_b)_{b\in[0,N]}$ with $s_0 = 0$ and $s_N = T - \delta$ satisfies for $k \in [0 : N-1]$, $s_{k+1} - s_k \le \kappa\left(1 \vee (T - s_{k+1})\right)$. Under Assumptions 4.3, 4.4, 4.5, and 4.6', we have the following error bound*

$$D_{\mathrm{KL}}(p_\delta \| q_{T-\delta}) \lesssim \exp(-\rho T)\log|\mathbb{X}| + \epsilon.$$

*Then with $T = \mathcal{O}\left(\frac{\log\left(\epsilon^{-1}\log|\mathbb{X}|\right)}{\rho}\right)$ and the early stopping scheme $\delta = \Omega(e^{-T})$, we have*

$$D_{\mathrm{KL}}(p_\delta \| q_{T-\delta}) \lesssim \epsilon \text{ with } \mathbb{E}[N] = \mathcal{O}\left(\frac{\overline{D}\log\left(\epsilon^{-1}\log|\mathbb{X}|\right)}{\rho}\right) \text{ steps.}$$

The proof of Theorem 4.9 is deferred to Appendix C, where a corresponding sketch is provided in Appendix C.1. Following a similar argument for Theorem 4.7, the dimensionality dependency of the uniformization scheme is $\widetilde{\mathcal{O}}(d)$, confirming the result for the special case $\mathbb{X} = \{0,1\}^d$ in (Chen & Ying, 2024) and $\mathbb{X} = [S]^d$ in (Zhang et al., 2024). Theorem 4.7 and 4.9 offer a direct comparison of the efficiency of the $\tau$-leaping and uniformization implementations for discrete diffusion models. Our proof reveals that the less favorable quadratic dependency in the $\tau$-leaping scheme arises from the truncation error, which is not present in the uniformization scheme, illustrating a possible advantage of the latter in reducing computational complexity.

Recalling the current state-of-the-art result for continuous diffusion models (Theorem 2.1) is $\widetilde{\mathcal{O}}(d)$, we conjecture that $\widetilde{\mathcal{O}}(d)$ is also the optimal rate in the discrete case. In the continuous case, the linear dependency does not depend on the accurate simulation of the approximate backward process (2.10) (or (3.2)) and is achievable with Euler-Maruyama schemes. The proof was via an intricate stochastic localization argument (Benton et al., 2023a), with which the bound on $\mathbb{E}[\|\nabla^2 \log p_t(x_t)\|^2]$ is improved by a $\mathcal{O}(d)$-factor from $\mathcal{O}(d^2)$. The corresponding argument for the $\tau$-leaping scheme of discrete diffusion models would be a possible refinement on Proposition C.2, which we believe is of independent interest and will be explored in future work.

## 5 CONCLUSION

In this paper, we have developed a comprehensive framework for the error analysis of discrete diffusion models. We rigorously introduced the Poisson random measure with evolving intensity and established the Lévy-type stochastic integral alongside change of measure arguments. These advancements not only hold mathematical significance but also facilitate a clear-cut analysis of discrete diffusion models. Moreover, we demonstrated that the inference process can be formulated as a stochastic integral using the Poisson random measure with evolving intensity, allowing the error to be systematically decomposed and optimized by algorithmic design, mirroring the theoretical framework for continuous diffusion models. Our framework unifies the error analysis of discrete diffusion models and provides the first error bounds for the $\tau$-leaping scheme in KL divergence.

Our results lay a theoretical groundwork for the design and analysis of discrete diffusion models, adaptable to broader contexts, such as time-inhomogeneous and non-symmetric rate matrices. Future research directions include exploring the continuum limit of discrete diffusion models in state spaces and time (Winkler et al., 2024; Zhao et al., 2024) and how to accelerate the implementation via parallel sampling (Chung et al., 2023; Shih et al., 2024; Tang et al., 2024; Cao et al., 2024; Selvam et al., 2024; Chen et al., 2024a; Gupta et al., 2024). We hope our work will inspire further research on both the theoretical analysis and practical applications of discrete diffusion models in various fields.

ACKNOWLEDGMENTS

Lexing Ying acknowledges the support of the National Science Foundation under Award No. DMS-2208163.

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

## A  MATHEMATICAL FRAMEWORK OF POISSON RANDOM MEASURE

In this section, we provide a mathematical framework for Poisson random measure with evolving intensity, which is crucial for the error analysis of discrete diffusion models in the main text.

### A.1  PRELIMINARIES

We first provide the definition of the ordinary Poisson random measure.

**Definition A.1** (Poisson Random Measure). *Let $(\Omega, \mathcal{F}, \mathbb{P})$ be a probability space and $(\mathbb{X}, \mathcal{B}, \nu)$ be a measure space satisfying that*

$$\int_{\mathbb{X}} 1 \vee |y| \vee |y|^2 \nu(\mathrm{d}y) < \infty,$$

*The random measure $N(\mathrm{d}t, \mathrm{d}y)$ on $\mathbb{R}^+ \times \mathbb{X}$ is called a Poisson random measure w.r.t. measure $\nu$ if it is a random counting measure satisfying the following properties:*

*(i) For any $B \in \mathcal{B}$ and $0 \leq s < t$, $N((s, t] \times B) \sim \mathcal{P}\left(\nu(B)(t - s)\right)$;*

*(ii) For any $t \geq 0$ and pairwise disjoint sets $\{B_i\}_{i \in [n]} \subset \mathcal{B}$, $\{N_t(B_i) := N((0, t] \times B_i)\}_{i \in [n]}$ are independent stochastic processes.*

The following definition of *predictability* will be frequently used for the well-definedness of stochastic integrals w.r.t. Poisson random measure, and thus the extension from ordinary Poisson random measure to Poisson random measure with evolving intensity.

**Definition A.2** (Predictability). *The predictable $\sigma$-algebra on $\mathbb{R}^+ \times \mathbb{X}$ is defined as the $\sigma$-algebra generated by all sets of the form $(s, t] \times B$ for $0 \leq s < t$ and $B \in \mathcal{B}$. A process $X_t$ is called predictable if and only if $X_t$ is predictable w.r.t. the predictable $\sigma$-algebra above.*

In the following, we will define the Poisson random measure with evolving intensity, which is a special case of random measures (Jacod & Shiryaev, 2013, Definition 1.3).

**Definition A.3** (Poisson Random Measure with Evolving Intensity). *Let $(\Omega, \mathcal{F}, \mathbb{P})$ be a probability space and $(\mathbb{X}, \mathcal{B}, \nu)$ be a measure space. Suppose $\lambda_t(y)$ is a non-negative predictable process on $\mathbb{R}^+ \times \mathbb{X} \times \Omega$ satisfying that for any $0 \leq T < \overline{T}$,*

$$\int_0^T \int_{\mathbb{X}} 1 \vee |y| \vee |y|^2 \lambda_t(y) \nu(\mathrm{d}y) \mathrm{d}t < \infty, \ a.s..$$

*The random measure $N[\lambda](\mathrm{d}t, \mathrm{d}y)$ on $\mathbb{R}^+ \times \mathbb{X}$ is called a Poisson random measure with evolving intensity $\lambda_t(y)$ w.r.t. measure $\nu$ if it is a random counting measure satisfying the following properties:*

*(i) For any $B \in \mathcal{B}$ and $0 \leq s < t$, $N[\lambda]((s, t] \times B) \sim \mathcal{P}\left(\int_s^t \int_B \lambda_\tau(y) \nu(\mathrm{d}y) \mathrm{d}\tau\right)$;*

*(ii) For any $t \geq 0$ and pairwise disjoint sets $\{B_i\}_{i \in [n]} \subset \mathcal{B}$,*

$$\{N_t[\lambda](B_i) := N[\lambda]((0, t] \times B_i)\}_{i \in [n]}$$

*are independent stochastic processes.*

**Theorem A.4** (Well-definedness of Poisson Random Measure with Evolving Intensity). *The Poisson random measure $N[\lambda](\mathrm{d}t, \mathrm{d}y)$ with evolving intensity $\lambda_t(y)$ is well-defined under the conditions in the definition above.*

*Proof.* We first augment the $(\mathbb{X}, \mathcal{B}, \nu)$ measure space to a product space $(\mathbb{X} \times \mathbb{R}, \mathcal{B} \times \mathcal{B}(\mathbb{R}), \nu \times m)$, where $m$ is the Lebesgue measure on $\mathbb{R}$, and $\mathcal{B}(\mathbb{R})$ is the Borel $\sigma$-algebra on $\mathbb{R}$. The Poisson random measure with evolving intensity $\lambda_t(y)$ can be defined in the augmented measure space as

$$N[\lambda]((s, t] \times B) := \int_s^t \int_B \int_{\mathbb{R}} \mathbf{1}_{0 \leq \xi \leq \lambda_\tau(y)} N(\mathrm{d}\tau, \mathrm{d}y, \mathrm{d}\xi), \tag{A.1}$$

where $N(\mathrm{d}\tau, \mathrm{d}y, \mathrm{d}\xi)$ is the Poisson random measure on $\mathbb{R}^+ \times \mathbb{X} \times \mathbb{R}$ w.r.t. measure $\nu(\mathrm{d}y)\mathrm{d}\xi$.

Then it is straightforward to verify the two conditions in the definition of Poisson random measure with evolving intensity by noticing that for pairwise disjoint sets $\{B_i\}_{i \in [n]} \subset \mathcal{B}$, $\{B_i \times \mathbb{R}\}_{i \in [n]} \subset \mathcal{B} \times \mathcal{B}(\mathbb{R})$ are also pairwise disjoint.

The Poisson random process $N[\lambda](\mathrm{d}t, \mathrm{d}y)$ with evolving intensity $\lambda_t(y)$ is well-defined up to an eventual explosion time

$$\overline{T} = \inf_T \left\{ \int_0^T \int_{\mathbb{X}} \lambda_t(y)\nu(\mathrm{d}y)\mathrm{d}t = \infty, \text{ a.s.} \right\}.$$

We refer the readers to (Protter, 1983) for a more rigorous detailed version of the proof. $\qquad\square$

**Remark A.5** (Relations to the classical Poisson random measure and that with state-dependent density). *The classical Poisson random measure is well-studied by the theory of Lévy processes (Applebaum, 2009), and the extension to the state-dependent intensity is proposed and analyzed in (Glasserman & Merener, 2004). Notably, Li (2007) establishes the stochastic integral formulation for the chemical master equation with the Poisson random measure with state-dependent intensity, which is a special case of the evolving intensity, and subsequently shows the weak and strong convergence of the $\tau$-leaping scheme.*

**Remark A.6** (Relation to the Cox process). *The Poisson random measure with evolving intensity shares multiple similarities with the Cox process (Cox, 1955; Last & Penrose, 2017), including being a point process and with the intensity being a random measure. The main difference is that the Cox process is defined on a general measure space, while the Poisson random measure with evolving intensity is defined on the product space $(\mathbb{X} \times \mathbb{R}, \mathcal{B} \times \mathcal{B}(\mathbb{R}), \nu \times m)$ and the intensity function is required to be predictable to ensure the well-definedness of its stochastic integral.*

### A.2 STOCHASTIC INTEGRAL W.R.T. POISSON RANDOM MEASURE

The following theorems provide the properties of stochastic integrals w.r.t. Poisson random measure with evolving intensity. The proofs are based on the observation that with the augmentation of the measure space argument (A.1), the stochastic integral w.r.t. Poisson random measure with evolving intensity in $(\mathbb{X}, \mathcal{B}, \nu)$ can be reduced to the stochastic integral w.r.t. homogeneous Poisson random measure in $(\mathbb{X} \times \mathbb{R}, \mathcal{B} \times \mathcal{B}(\mathbb{R}), \nu \times m)$, and under certain conditions on the measure space $(\mathbb{X}, \mathcal{B}, \nu)$, to the well-known Lévy-type stochastic integral (Applebaum, 2009). For simplicity, we will work on the interval $t \in [0, T]$ with $T < \overline{T}$ and the following regularity conditions of the Poisson random measure:

$$0 < \operatorname*{ess\,inf}_{\tau \in [0,T], y \in \mathbb{X}} \lambda_\tau(y) \leq \operatorname*{ess\,sup}_{\tau \in [0,T], y \in \mathbb{X}} \lambda_\tau(y) < +\infty.$$

One can easily generalize the following results to their local versions on $[0, \overline{T})$ by considering its compact subsets.

**Theorem A.7** (Stochastic Integrals w.r.t. Poisson Random Measure with Evolving Density). *For any predictable process $K_t(y)$ on $\mathbb{R}^+ \times \mathbb{X} \times \Omega$, the stochastic integral w.r.t. Poisson random measure with evolving intensity $\lambda_t(y)$*

$$x_t = x_0 + \int_0^t \int_{\mathbb{X}} K_\tau(y) N[\lambda](\mathrm{d}\tau, \mathrm{d}y), \tag{A.2}$$

*has a unique solution, for which the following properties hold:*

*(1) (Expectation) For any $t \geq 0$, we have*

$$\mathbb{E}\left[ \int_0^t \int_{\mathbb{X}} K_\tau(y) N[\lambda](\mathrm{d}\tau, \mathrm{d}y) \right] = \int_0^t \int_{\mathbb{X}} K_\tau(y) \lambda_\tau(y) \nu(\mathrm{d}y) \mathrm{d}\tau;$$

*(2) (Martingale) For any $t \geq 0$, we have*

$$\int_0^t \int_{\mathbb{X}} K_\tau(y) \widetilde{N}[\lambda](\mathrm{d}\tau, \mathrm{d}y) := \int_0^t \int_{\mathbb{X}} K_\tau(y) N[\lambda](\mathrm{d}\tau, \mathrm{d}y) - \int_0^t \int_{\mathbb{X}} K_\tau(y) \lambda_\tau(y) \nu(\mathrm{d}y) \mathrm{d}\tau$$

*is a local $\mathcal{F}_t$-martingale;*

*(3) (Itô Isometry) For any $t \geq 0$, we have*

$$\mathbb{E}\left[\left(\int_0^t \int_{\mathbb{X}} K_\tau(y) N[\lambda](\mathrm{d}\tau, \mathrm{d}y)\right)^2\right] = \int_0^t \int_{\mathbb{X}} K_\tau(y)^2 \lambda_\tau(y) \nu(\mathrm{d}y) \mathrm{d}\tau.$$

*Proof.* We first write the integral (A.2) in the augmented measure space $(\mathbb{X} \times \mathbb{R}, \mathcal{B} \times \mathcal{B}(\mathbb{R}), \nu \times m)$ as

$$x_t = x_0 + \int_0^t \int_{\mathbb{X}} \int_{\mathbb{R}} K_\tau(y) \mathbf{1}_{0 \leq \xi \leq \lambda_\tau(y)} N(\mathrm{d}\tau, \mathrm{d}y, \mathrm{d}\xi), \tag{A.3}$$

and since $K_t(y)\mathbf{1}_{0 \leq \xi \leq \lambda_t(y)}$ is a predictable process, the desired properties can be derived from the corresponding properties of the stochastic integral w.r.t. Poisson random measure in the augmented measure space.

The subsequent proof will follow a similar argument as the proof of the stochastic integral w.r.t. Brownian motion (*e.g.* in (Øksendal, 2003)) by starting from proving the properties for elementary processes, which in our case refer to working with the *elementary predictable processes* of the following form:

$$Z_t(y, \xi)(\omega) = \sum_{i=0}^{n-1} \sum_{j=1}^{m} \sum_{k=1}^{l} Z_{i,j,k}(\omega) \mathbf{1}_{t \in (t_i, t_{i+1}]} \mathbf{1}_{y \in B_j} \mathbf{1}_{\xi \in C_k},$$

where $0 = t_0 < \cdots < t_n = T$ is a partition of $[0, T]$, $B_j \in \mathcal{B}$ for $j \in [m]$ are a partition of $\mathbb{X}$ with $\nu(B_k) < \infty$, and $C_k \in \mathcal{B}(\mathbb{R})$ for $k \in [l]$ are a partition of the time interval $[0, \mathrm{ess\,sup}_{\tau \in [0,T], y \in \mathbb{X}} \lambda_\tau(y)]$ with $m(C_k) < \infty$, and $K_{i,j,k}$ is bounded and $\mathcal{F}_{t_i}$-measurable, on which the stochastic integral is defined as

$$\int_0^t \int_{\mathbb{X}} Z_\tau(y, \xi) N(\mathrm{d}\tau, \mathrm{d}y, \mathrm{d}\xi) = \sum_{i=0}^{n-1} \sum_{j=1}^{m} \sum_{k=1}^{l} Z_{i,j,k} N_{t_i}((t_i, t_{i+1}] \times B_j \times C_k).$$

Then, it is straightforward to verify the properties of the stochastic integral for the elementary predictable process $Z_t^+(y, \xi)$, using the definition of Poisson random measure (Definition A.1). For general predictable processes $Z_t(y, \xi)$, we write $Z_t(y, \xi) = Z_t^+(y, \xi) - Z_t^-(y, \xi)$, where $Z_t^+(y, \xi)$ and $Z_t^-(y, \xi)$ are positive and negative parts of $Z_t(y, \xi)$, and apply the results to $Z_t^+(y, \xi)$ and $Z_t^-(y, \xi)$ separately.

Finally, we take $Z_t(y, \xi) = K_t(y)\mathbf{1}_{0 \leq \xi \leq \lambda_t(y)}$ to derive the properties of the stochastic integral w.r.t. Poisson random measure with evolving intensity.

We refer readers to (Eberle, 2015, Section 2.2) for detailed arguments. For the uniqueness of the solution to the stochastic integral, we also refer to (Protter, 1983, Theorem 3.1). $\qquad\square$

**Proposition A.8.** *Define the list of jump times $(t_n)_{n \in \mathbb{N}}$ recursively as*

$$t_0 = 0, \quad t_{n+1} = \inf\{t > t_n | \Delta x_t \neq 0\}, \ n \geq 0,$$

*the Poisson random measure $N[\lambda](\mathrm{d}t, \mathrm{d}y)$ with evolving intensity $\lambda_t(y)$ can be written as*

$$N[\lambda](\mathrm{d}t, \mathrm{d}y) = \sum_{n=1}^{\infty} \delta_{t_n}(\mathrm{d}t) \delta_{Y_n}(\mathrm{d}y), \tag{A.4}$$

*and the stochastic integral (A.2) is càdlàg and can be rewritten as a sum of jumps:*

$$x_t = x_0 + \sum_{n=1}^{N} \Delta x_{t_n} = x_0 + \sum_{n=1}^{N} K_{t_n}(Y_n), \tag{A.5}$$

*where $N$ is a random variable satisfying $t_N \leq t < t_{N+1}$, and $\Delta x_{t_n}$ are the jumps $\Delta x_{t_n} = x_{t_n} - x_{t_n^-}$ with $x_{t_n^-} := \lim_{s \to t_n^-} x_s$.*

*Proof.* To see the solution is càdlàg, we notice the following right limit at time $t$:

$$\lim_{\epsilon \to 0} (x_{t+\epsilon} - x_t) = \int_{(t,t+\epsilon] \times \mathbb{X}} K_t(y) N[\lambda](\mathrm{d}t, \mathrm{d}y) \to 0,$$

and the left limit at time $t$:

$$\Delta x_t = \lim_{\epsilon \to 0} (x_t - x_{t-\epsilon}) = \int_{(t-\epsilon,t] \times \mathbb{X}} K_t(y) N[\lambda](\mathrm{d}t, \mathrm{d}y) \to \int_{\mathbb{X}} K_t(y) N[\lambda](\{t\} \times \mathrm{d}y), \quad \text{(A.6)}$$

where the notation $N[\lambda](\{t\} \times \mathrm{d}y)$ should be understood as $N[\lambda](\{t\} \times \mathrm{d}y) = 0$ if $t \notin \{t_n\}_{n\in\mathbb{N}}$, or otherwise $Y_n = N[\lambda](\{t_n\} \times \mathrm{d}y)$ is a random variable on $\mathbb{X}$.

Since the Poisson random measure $N[\lambda](\mathrm{d}t, \mathrm{d}y)$ with evolving intensity $\lambda_t(y)$ is a random counting measure, it can be represented as a countable sum of Dirac measures as in (A.4), and thus we have

$$x_t = x_0 + \int_0^t \int_{\mathbb{X}} K_t(y) N[\lambda](\mathrm{d}t, \mathrm{d}y)$$

$$= x_0 + \int_0^t \int_{\mathbb{X}} K_t(y) \sum_{n=1}^N \delta_{t_n}(\mathrm{d}t) \delta_{Y_n}(\mathrm{d}y) = x_0 + \sum_{n=1}^N K_{t_n}(Y_n).$$

By the definition of Poisson random measure, $(t_n)_{n\in[N]}$ are also the jump times of the homogeneous Poisson random measure $N(\mathrm{d}t, \mathrm{d}y, \mathrm{d}\xi)$ in the augmented measure space $(\mathbb{X} \times \mathbb{R}, \mathcal{B} \times \mathcal{B}(\mathbb{R}), \nu \times m)$ w.r.t. measure $\nu(\mathrm{d}y)\mathrm{d}\xi$. Therefore, with a slight abuse of notations, we will assume $(t_n, Y_n, \Xi_n)_{n\in[N]}$ are i.i.d. random variables with probability measure proportional to $\mathrm{d}t\nu(\mathrm{d}y)\mathrm{d}\xi$, for each of which $\Xi_n \leq \lambda_{t_n}(Y_n)$ holds because otherwise the jump would not occur.

Then the distribution of $Y_n$ can be derived as a conditional probability of the jump location $Y_n$ given the jump time $t_n$ and $\Xi_n \leq \lambda_{t_n}(Y_n)$:

$$\mathbb{P}(Y_n = y)\nu(\mathrm{d}y) = \frac{\int_{\mathbb{R}} \nu(\mathrm{d}y) \mathbf{1}_{\Xi_n \leq \lambda_{t_n}(y)} \mathrm{d}\xi}{\int_{\mathbb{R}} \int_{\mathbb{X}} \nu(\mathrm{d}y) \mathbf{1}_{\Xi_n \leq \lambda_{t_n}(y)} \mathrm{d}\xi} = \frac{\lambda_{t_n}(y)\nu(\mathrm{d}y)}{\int_{\mathbb{X}} \lambda_{t_n}(y)\nu(\mathrm{d}y)}, \quad \text{(A.7)}$$

and the proof is complete. $\qquad \square$

The following theorem gives the martingale characterization of Poisson random measure with evolving intensity, which will be crucial for the proof of the change of measure arguments:

**Theorem A.9** (Martingale Characterization of Poisson Random Measure with Evolving Density)**.** *Let $N[\lambda](\mathrm{d}t, \mathrm{d}y)$ be a $\mathcal{F}_t$-adapted process in the probability space $(\Omega, \mathcal{F}, \mathbb{P})$. Then $N[\lambda](\mathrm{d}t, \mathrm{d}y)$ is a Poisson random measure with evolving intensity $\lambda_t(y)$ if and only if the complex-valued process*

$$M_t[f] = \exp\left( i \int_0^t \int_{\mathbb{X}} f_\tau(y) N[\lambda](\mathrm{d}\tau, \mathrm{d}y) + \int_0^t \int_{\mathbb{X}} \left( 1 - e^{if_\tau(y)} \right) \lambda_\tau(y)\nu(\mathrm{d}y)\mathrm{d}\tau \right) \quad \text{(A.8)}$$

*is a local martingale for any predictable process $f_\tau(y)$ satisfying that $f_\tau(y) \in L^1(\mathbb{X}, \nu)$, a.s..*

*Proof.* By Proposition A.8, we rewrite the stochastic integral as a sum of jumps:

$$\int_0^t \int_{\mathbb{X}} f_t(y) N[\lambda](\mathrm{d}\tau, \mathrm{d}y) = \sum_{n=1}^N f_{t_n}(Y_n),$$

where $(t_n, Y_n, \Xi_n)_{n\in[N]}$ are i.i.d. random variables with probability measure proportional to $\mathrm{d}t\nu(\mathrm{d}y)\mathrm{d}\xi$, for each of which $\Xi_n \leq \lambda_{t_n}(Y_n)$ holds, following a similar argument as in the proof of Proposition A.8.

Then, it is straightforward to derive the following probability of the jump time $t_n = \tau$:

$$\mathbb{P}(t_n = \tau)\mathrm{d}\tau = \frac{\int_{\mathbb{X}} \mathbb{P}(Y_n = y, t_n = \tau)\nu(\mathrm{d}y)\mathrm{d}\tau}{\int_0^t \int_{\mathbb{X}} \mathbb{P}(Y_n = y, t_n = \tau)\nu(\mathrm{d}y)\mathrm{d}\tau} = \frac{\int_{\mathbb{X}} \lambda_\tau(y)\nu(\mathrm{d}y)\mathrm{d}\tau}{\int_0^t \int_{\mathbb{X}} \lambda_\tau(y)\nu(\mathrm{d}y)\mathrm{d}\tau};$$

and by the definition of the Poisson random measure, we have the following probability of the total number of jumps $N = n$:

$$\mathbb{P}(N = n) = \frac{1}{n!} \exp\left(-\int_0^t \int_{\mathbb{X}} \lambda_\tau(y)\nu(\mathrm{d}y)\mathrm{d}\tau\right)\left(\int_0^t \int_{\mathbb{X}} \lambda_\tau(y)\nu(\mathrm{d}y)\mathrm{d}\tau\right)^n.$$

Without loss of generality, we only verify $\mathbb{E}[M_t[f]] = 1$ as follows, and general cases are similar by Markov property:

$$\mathbb{E}\left[\exp\left(i\int_0^t \int_{\mathbb{X}} f_{t_n}(y)N[\lambda](\mathrm{d}\tau, \mathrm{d}y)\right)\right]$$

$$=\mathbb{E}\left[\exp\left(i\sum_{n=1}^N f_{t_n}(Y_n)\right)\Big|\Xi_n \leq \lambda_{t_n}(Y_n),\ \forall n \in [N]\right]$$

$$=\mathbb{E}\left[\prod_{n=1}^N \mathbb{E}\left[e^{if_{t_n}(Y_n)}\big|\Xi_n \leq \lambda_{t_n}(Y_n)\right]\right]$$

$$=\mathbb{E}\left[\prod_{n=1}^N \mathbb{E}\left[\int_{\mathbb{X}} \frac{e^{if_\tau(y)}\lambda_\tau(y)\nu(\mathrm{d}y)}{\int_{\mathbb{X}} \lambda_\tau(y)\nu(\mathrm{d}y)}\Big|t_n = \tau\right]\right]$$

$$=\sum_{n=1}^\infty \frac{1}{n!}\left(\int_0^t \int_{\mathbb{X}} e^{if_\tau(y)}\lambda_\tau(y)\nu(\mathrm{d}y)\mathrm{d}\tau\right)^n \exp\left(-\int_0^t \int_{\mathbb{X}} \lambda_\tau(y)\nu(\mathrm{d}y)\mathrm{d}\tau\right)$$

$$= \exp\left(\int_0^t \int_{\mathbb{X}} \left(e^{if_\tau(y)} - 1\right)\lambda_\tau(y)\nu(\mathrm{d}y)\mathrm{d}\tau\right),$$

which immediately yields the desired result $\mathbb{E}[M_t[f]] = 1$.

On the other hand, for any $0 \leq s < t$ and $B \in \mathcal{B}$, we set

$$Z_t(y) = u\mathbf{1}_{t \in (s,t]}\mathbf{1}_{y \in B},$$

where $u \in \mathbb{R}$, and by assumption, we have

$$\mathbb{E}[M_t[Z]] = \mathbb{E}\left[\exp\left(i\int_0^t \int_{\mathbb{X}} Z_\tau(y)N[\lambda](\mathrm{d}\tau, \mathrm{d}y) + \int_0^t \int_{\mathbb{X}} \left(1 - e^{iZ_\tau(y)}\right)\lambda_\tau(y)\nu(\mathrm{d}y)\mathrm{d}\tau\right)\right]$$

$$=\mathbb{E}\left[\exp\left(iu\int_s^t \int_B N[\lambda](\mathrm{d}\tau, \mathrm{d}y) + \int_s^t \int_{\mathbb{X}} \left(1 - e^{iu}\right)\lambda_\tau(y)\nu(\mathrm{d}y)\mathrm{d}\tau\right)\right]$$

$$=\mathbb{E}\left[\exp\left(iuN[\lambda]((s,t] \times B) + \left(1 - e^{iu}\right)(t-s)\int_{\mathbb{X}} \lambda_\tau(y)\nu(\mathrm{d}y)\right)\right] = 1,$$

*i.e.* the following holds for any $u \in \mathbb{R}$:

$$\mathbb{E}\left[\exp\left(iuN[\lambda]((s,t] \times B))\right)\right] = \left(e^{iu} - 1\right)(t-s)\int_{\mathbb{X}} \lambda_\tau(y)\nu(\mathrm{d}y),$$

which by Lévy's continuity theorem implies that

$$N[\lambda]((s,t] \times B) \sim \mathcal{P}\left((t-s)\int_{\mathbb{X}} \lambda_\tau(y)\nu(\mathrm{d}y)\right),$$

and thus $N[\lambda](\mathrm{d}t, \mathrm{d}y)$ is a Poisson random measure with evolving intensity $\lambda_t(y)$ by Definition A.1. $\square$

**Theorem A.10** (Itô's Formula for Poisson Random Measure with Evolving Density). *Let $N[\lambda](\mathrm{d}t, \mathrm{d}y)$ be a Poisson random measure with evolving intensity $\lambda_t(y)$ in the probability space $(\Omega, \mathcal{F}, \mathbb{P})$ and $K_t(y)$ be a predictable process on $\mathbb{R}^+ \times \mathbb{X} \times \Omega$. Suppose a process $x_t$ satisfies the stochastic integral*

$$x_t = x_0 + \int_0^t \int_{\mathbb{X}} K_\tau(y)N[\lambda](\mathrm{d}\tau, \mathrm{d}y), \tag{A.9}$$

then for any $f_t(y) \in C(\mathbb{R}^+ \times \mathbb{X})$ with probability 1, we have

$$f_t(x_t) = f_0(x_0) + \int_0^t \partial_\tau f_\tau(x_\tau) \mathrm{d}\tau + \int_0^t \int_{\mathbb{X}} (f_\tau(x_{\tau^-} + K_\tau(y)) - f_\tau(x_{\tau^-})) \, N[\lambda](\mathrm{d}\tau, \mathrm{d}y).$$

*Proof.* By Proposition A.8, we again rewrite the stochastic integral as a sum of jumps:

$$x_t = x_0 + \sum_{n=1}^N K_{t_n}(Y_n),$$

where $(t_n)_{n \in [N]}$ are the jump times with $t_0 = 0$ and $t_N \le t < t_{N+1}$, and $(Y_n)_{n \in [N]}$ are the jump locations. Consequently, it is easy to see that $x_{t_n^-} = x_t = x_{t_{n-1}}$ for $t \in (t_{n-1}, t_n]$ and $n \in [N]$.

Then we have the following decomposition:

$$f_t(x_t) - f_0(x_0)$$

$$= f_t(x_t) - f_{t_N}(x_{t_N}) + \sum_{n=1}^N \left( f_{t_n}(x_{t_n}) - f_{t_n}(x_{t_n^-}) + f_{t_n}(x_{t_n^-}) - f_{t_{n-1}}(x_{t_{n-1}}) \right)$$

$$= \int_{t_N}^t \partial_\tau f_\tau(x_{t_N}) \mathrm{d}\tau + \sum_{n=1}^N \int_{t_{n-1}}^{t_n} \partial_\tau f_\tau(x_{t_{n-1}}) \mathrm{d}\tau + \sum_{n=1}^N \left( f_{t_n}(x_{t_{n-1}} + K_{t_n}(Y_n)) - f_{t_n}(x_{t_{n-1}}) \right),$$

and for the last term in the above equation, we have

$$\sum_{n=1}^N \left( f_{t_n}(x_{t_{n-1}} + K_{t_n}(Y_n)) - f_{t_n}(x_{t_{n-1}}) \right) = \sum_{n=1}^N \left( f_{t_n}(x_{t_n^-} + K_{t_n}(Y_n)) - f_{t_n}(x_{t_n^-}) \right)$$

$$= \int_0^t \int_{\mathbb{X}} (f_\tau(x_{\tau^-} + K_\tau(y)) - f_\tau(x_{\tau^-})) \, N[\lambda](\mathrm{d}\tau, \mathrm{d}y).$$

Combining the above results, we have the desired result. $\square$

**Lemma A.11.** *Denote the trajectory obtained by simulating the master equation (2.7) of the forward process of the discrete diffusion model as $x_t$, then the time interval $\Delta t_n = t_{n+1} - t_n$ is distributed according to the following distribution:*

$$\mathbb{P}(\Delta t_n > \tau) = \exp\left( \int_0^\tau Q_{\tau'}(x_{t_n}, x_{t_n}) \mathrm{d}\tau' \right), \tag{A.10}$$

*and the jump location $x_{t_{n+1}}$ is distributed according to the following distribution:*

$$\mathbb{P}(x_{t_{n+1}} = y) = -\frac{Q_{t_{n+1}}(y, x_{t_n})}{Q_{t_{n+1}}(x_{t_n}, x_{t_n})}. \tag{A.11}$$

*Proof.* The results can be found in (Eberle, 2009, Section 1.2). While a fully rigorous proof can be conducted by discretizing the time-inhomogeneous continuous-time Markov chain into $2^n$ uniform steps and taking the limit as $n \to \infty$, following the approach in (Lalley, 2012), we will provide a more intuitive proof here for completeness.

Set $\boldsymbol{p}_{t_n} = \boldsymbol{e}_{x_{t_n}}$, where $\boldsymbol{e}_y$ is the $y$-th unit vector in $\mathbb{R}^{|\mathbb{X}|}$. then the $x_{t_n}$-th entry of (2.7) yields

$$\frac{\mathrm{d}}{\mathrm{d}t} \mathbb{P}(x_t = x_{t_n}) = \frac{\mathrm{d}}{\mathrm{d}t} p_t(x_{t_n}) = \sum_{y \in \mathbb{X}} Q_t(x_{t_n}, y) p_t(y),$$

which, by the assumed continuity of the rate matrix $\boldsymbol{Q}_t$, implies

$$\mathbb{P}(\Delta t_n > \tau) = \mathbb{P}(x_{t_n + \tau} = x_{t_n}) = 1 + Q_{t_n}(x_{t_n}, x_{t_n}) \tau + o(\tau),$$

and thus

$$\frac{\mathrm{d}}{\mathrm{d}\tau} \log \mathbb{P}(\Delta t_n > \tau) = \lim_{\tau \to 0} \frac{\log \mathbb{P}(\Delta t_n > \tau)}{\tau} = \lim_{\tau \to 0} Q_{t_n}(x_{t_n}, x_{t_n}) + o(1) = Q_{t_n}(x_{t_n}, x_{t_n}),$$

integrating the above equation yields the desired result.

Similarly, by setting $\boldsymbol{p}_{t_{n+1}-\tau} = \boldsymbol{e}_{x_{t_n}}$, we have for all $y \in \mathbb{X} \setminus \{x_{t_n}\}$:

$$\mathbb{P}(x_{t_{n+1}} = y) = \boldsymbol{p}_{t_{n+1}}(y)$$
$$= \lim_{\tau \to 0} \frac{\boldsymbol{p}_{t_{n+1}-\tau}(y) + Q_{t_{n+1}-\tau}(y, x_{t_n})\tau + o(\tau)}{\sum_{y \in \mathbb{X} \setminus \{x_{t_n}\}} \left(\boldsymbol{p}_{t_{n+1}-\tau}(y) + Q_{t_{n+1}-\tau}(y, x_{t_n})\tau + o(\tau)\right)}$$
$$= \frac{Q_{t_{n+1}}(y, x_{t_n})}{\sum_{y \in \mathbb{X} \setminus \{x_{t_n}\}} Q_{t_{n+1}}(y, x_{t_n})} = -\frac{Q_{t_{n+1}}(y, x_{t_n})}{Q_{t_{n+1}}(x_{t_n}, x_{t_n})},$$

and the result follows. $\square$

### A.3 PROOFS OF CHANGE OF MEASURE RELATED ARGUMENTS

*Proof of Theorem 3.3.* In the following, we will denote the expectation under the measure $\mathbb{P}$ by $\mathbb{E}_{\mathbb{P}}$ and the expectation under the measure $\mathbb{Q}$ by $\mathbb{E}_{\mathbb{Q}}$.

By Theorem A.9, to verify that the Poisson random measure $N[\lambda](\mathrm{d}t, \mathrm{d}y)$ with evolving intensity $\lambda_t(y)$ is a Poisson random measure with evolving intensity $\lambda_t(y)h_t(y)$ under the measure $\mathbb{Q}$, it suffices to show that for any $f \in L^1(\mathbb{X}, \nu)$, the complex-valued process

$$M_t[f] = \exp\left(i\int_0^t \int_{\mathbb{X}} f(y)N[\lambda](\mathrm{d}\tau, \mathrm{d}y) + \int_0^t \int_{\mathbb{X}} \left(1 - e^{if(y)}\right)\lambda_\tau(y)h_\tau(y)\nu(\mathrm{d}y)\mathrm{d}\tau\right)$$

is a local martingale under the measure $\mathbb{Q}$.

To this end, we perform the following calculation:

$$\mathbb{E}_{\mathbb{Q}}[M_t[f]] = \mathbb{E}_{\mathbb{P}}[M_t[f]Z_t[h]]$$
$$= \mathbb{E}_{\mathbb{P}}\left[\exp\left(i\int_0^t \int_{\mathbb{X}} f(y)N[\lambda](\mathrm{d}\tau, \mathrm{d}y) + \int_0^t \int_{\mathbb{X}} \left(1 - e^{if(y)}\right)\lambda_\tau(y)h_\tau(y)\nu(\mathrm{d}y)\mathrm{d}\tau\right)\right.$$
$$\left.\exp\left(\int_0^t \int_{\mathbb{X}} \log h_t(y)N[\lambda](\mathrm{d}t \times \mathrm{d}y) - \int_0^t \int_{\mathbb{X}} (h_t(y) - 1)\lambda_t(y)\nu(\mathrm{d}y)\right)\right]$$
$$= \mathbb{E}_{\mathbb{P}}\left[\exp\left(i\int_0^t \int_{\mathbb{X}} (f(y) + \log h_t(y))N[\lambda](\mathrm{d}\tau, \mathrm{d}y) + \int_0^t \int_{\mathbb{X}} \left(1 - e^{if(y)}h_t(y)\right)\lambda_t(y)\nu(\mathrm{d}y)\right)\right]$$
$$= \mathbb{E}_{\mathbb{P}}[M_t[f + \log h]],$$

and by assumption, $f + \log h \in L^1(\mathbb{X}, \nu)$, a.s., which implies that $M_t[f + \log h]$ is a local martingale under the measure $\mathbb{P}$ again by Theorem A.9. Consequently, $M_t[f]$ is a local martingale under the measure $\mathbb{Q}$, and the result follows. $\square$

**Remark A.12.** *One may verify that the exponential process $Z_t$ in (3.3) is a local $\mathcal{F}_t$-martingale by Applebaum (2009, Corollary 5.2.2) under mild assumptions on the function $h_t(y)$ (cf. Novikov's condition in the Girsanov's theorem for Itô integrals).*

*Proof of Corollary 3.4.* With a similar argument as in the proof of Proposition 3.2, we have the following stochastic integral representation of the approximate backward process with the learned neural network score function $\widehat{s}_t^\theta$:

$$y_s = y_0 + \int_0^s \int_{\mathbb{X}} (y - y_{s-})N[\widehat{\mu}^\theta](\mathrm{d}s, \mathrm{d}y), \text{ with } \widehat{\mu}_s^\theta(y) = \overleftarrow{\widehat{s}}_s^\theta(y_{s-}, y)\widetilde{Q}_s(y_{s-}, y) = \widehat{\mu}_s^\theta(y). \quad \text{(A.12)}$$

By the data-processing inequality and the chain rule of KL divergence, we have

$$D_{\mathrm{KL}}(\breve{p}_T \| q_T) \le D_{\mathrm{KL}}(\breve{p}_{0:T} \| q_{0:T}) = D_{\mathrm{KL}}(\breve{p}_0 \| q_0) + \mathbb{E}\left[D_{\mathrm{KL}}(\breve{p}_{0:T} \| q_{0:T} | \breve{x}_0 = y_0 = y)\right].$$

Then notice that conditioning on the alignment of the initial state $\breve{x}_0 = y_0 = y$ for any $y \in \mathbb{X}$, the second term in the above equation can be expressed as

$$D_{\mathrm{KL}}(\breve{p}_{0:T} \| q_{0:T} | \breve{x}_0 = y_0 = y) = \mathbb{E}\left[\log \frac{\mathrm{d}\breve{p}_{0:T}}{\mathrm{d}q_{0:T}} \bigg| \breve{x}_0 = y_0 = y\right] = \mathbb{E}\left[\log Z_T^{-1}\left[\frac{\widehat{\mu}^\theta}{\mu}\right]\right],$$

where the last equality is by the change of measure in Theorem 3.3 from the stochastic integral formulation (3.2) of the backward process (2.8) with the true score function $\breve{s}$ to the stochastic integral formulation (A.12) of the approximate backward process with the learned score function $\widehat{s}^\theta$.

Plug in the expression of $Z_T$ in (3.3) and notice that

$$\frac{\widehat{\mu}_s^\theta}{\mu_s} = \frac{\widehat{\breve{s}}_s^\theta(y_{s-}, y)\widetilde{Q}_s(y_{s-}, y)}{\breve{s}_s(y_{s-}, y)\widetilde{Q}_s(y_{s-}, y)} = \frac{\widehat{\breve{s}}_s^\theta(y_{s-}, y)}{\breve{s}_s(y_{s-}, y)},$$

we have

$$\mathbb{E}\left[\log Z_T^{-1}\left[\frac{\widehat{\mu}^\theta}{\mu}\right]\right]$$

$$= \mathbb{E}\left[-\int_0^T \int_{\mathbb{X}} \log \frac{\widehat{\breve{s}}_s^\theta(y_{s-}, y)}{\breve{s}_s(y_{s-}, y)} N[\mu](\mathrm{d}s \times \mathrm{d}y) + \int_0^T \int_{\mathbb{X}} \left(\frac{\widehat{\breve{s}}_s^\theta(y_{s-}, y)}{\breve{s}_s(y_{s-}, y)} - 1\right)\mu_s(y)\nu(\mathrm{d}y)\mathrm{d}s\right]$$

$$= \mathbb{E}\left[\int_0^T \int_{\mathbb{X}} \left(\frac{\widehat{\breve{s}}_s^\theta(y_{s-}, y)}{\breve{s}_s(y_{s-}, y)} - 1 - \log \frac{\widehat{\breve{s}}_s^\theta(y_{s-}, y)}{\breve{s}_s(y_{s-}, y)}\right)\breve{s}_s(y_{s-}, y)\widetilde{Q}_s(y_{s-}, y)\nu(\mathrm{d}y)\mathrm{d}s\right]$$

$$= \mathbb{E}\left[\int_0^T \int_{\mathbb{X}} \left(\widehat{\breve{s}}_s^\theta(y_{s-}, y) - \breve{s}_s(y_{s-}, y) - \breve{s}_s(y_{s-}, y)\log \frac{\widehat{\breve{s}}_s^\theta(y_{s-}, y)}{\breve{s}_s(y_{s-}, y)}\right)\widetilde{Q}_s(y_{s-}, y)\nu(\mathrm{d}y)\mathrm{d}s\right],$$

rearranging the terms in the above equation yields the desired result. $\qquad\square$

## A.4 PROOFS OF STOCHASTIC INTEGRAL FORMULATIONS

*Proof of Proposition 3.2.* In the following, we will denote the trajectory obtained by simulating the master equation (2.7) of the forward process of the discrete diffusion model as $\widetilde{x}_t$ and the trajectory obtained by the stochastic integral (3.1) as $x_t$, with $x_0 = \widetilde{x}_0$. We will also use the notation $\widetilde{\cdot}$ to denote the quantities associated with the trajectory $\widetilde{x}_t$. The goal is to show that $x_t$ and $\widetilde{x}_t$ are identically distributed for any $t \in [0, T]$.

We prove this claim by induction. We assume that for any $t \in [0, t_n]$, where $n \in \mathbb{N}$ and $t_n$ is the $n$-th jump time with $t_0 = 0$, the two trajectories $x_t$ and $\widetilde{x}_t$ are identically distributed. For simplicity, we realign the two processes $x_t$ and $\widetilde{x}_t$ at time $t_n$ by setting $x_{t_n} = \widetilde{x}_{t_n}$.

We first consider the process $\widetilde{x}_t$ generated by the discrete diffusion model (2.7). Recall the definition $\lambda_t(y) = \widetilde{Q}_t(y, \widetilde{x}_{t-})$, we have that

$$\int_{\mathbb{X}} \lambda_t(y)\nu(\mathrm{d}y) = \sum_{y \in \mathbb{X}} \widetilde{Q}_t(y, \widetilde{x}_{t-}) = -Q_t(\widetilde{x}_{t-}, \widetilde{x}_{t-}) = -Q_t(\widetilde{x}_{t_n}, \widetilde{x}_{t_n}), \text{ for } t \in (t_n, t_{n+1}].$$

By Lemma A.11, the time interval $\Delta\widetilde{t}_n = \widetilde{t}_{n+1} - \widetilde{t}_n$ is distributed according to (A.10), *i.e.*

$$\mathbb{P}(\Delta\widetilde{t}_n > \tau) = \exp\left(\int_0^\tau Q_{\tau'}(\widetilde{x}_{t_n}, \widetilde{x}_{t_n})\mathrm{d}\tau'\right) = \exp\left(-\int_0^\tau \int_{\mathbb{X}} \lambda_{\tau'}(y)\nu(\mathrm{d}y)\mathrm{d}\tau'\right).$$

Similarly, the jump location $\widetilde{x}_{t_{n+1}}$ is distributed according to (A.11), *i.e.*

$$\mathbb{P}(\widetilde{x}_{t_{n+1}} = y) = -\frac{Q_{t_{n+1}}(y, \widetilde{x}_{t_n})}{Q_{t_{n+1}}(\widetilde{x}_{t_n}, \widetilde{x}_{t_n})} = \frac{\lambda_{t_{n+1}}(y)}{\int_{\mathbb{X}} \lambda_{t_{n+1}}(y)\nu(\mathrm{d}y)}.$$

Now we turn to the stochastic integral (3.1). By definition of the Poisson random measure, we have

$$\mathbb{P}(\Delta t_n > \tau) = \mathbb{P}(N[\lambda]((t_n, t_n + \tau] \times \mathbb{X}) = 0) = \exp\left(-\int_{t_n}^{t_n+\tau} \int_{\mathbb{X}} \lambda_{\tau'}(y)\nu(\mathrm{d}y)\mathrm{d}\tau'\right), \quad \text{(A.13)}$$

and the jump location is distributed according to (A.14), *i.e.*

$$\mathbb{P}(x_{t_{n+1}} = y) = \frac{\lambda_{t_{n+1}}(y)}{\int_{\mathbb{X}} \lambda_{t_{n+1}}(y)\nu(\mathrm{d}y)}. \tag{A.14}$$

Comparing the arguments above, we conclude that the two processes $x_t$ and $\widetilde{x}_t$ are identically distributed for any $t \in [0, t_{n+1}]$, and the induction is complete.

The proof of the equivalence between the backward process of the discrete diffusion model governed by (2.8) and the corresponding stochastic integral (3.2) can be conducted similarly, and the result follows. □

### A.4.1 $\tau$-LEAPING

*Proof of Proposition 4.1.* Without loss of generality, we give the proof for $s = s_N$, and the general case can be proved similarly.

The stochastic integral (4.2) can be partitioned by the time discretization $(s_i)_{i \in [0:N]}$ into $N$ intervals along which the evolving intensity is constant, *i.e.*

$$\widehat{y}_{s_N} = \widehat{y}_0 + \int_0^s \int_{\mathbb{X}} (y - \widehat{y}_{\lfloor s \rfloor^-})N[\widehat{\mu}_{\lfloor \cdot \rfloor}](\mathrm{d}s, \mathrm{d}y)$$

$$= \widehat{y}_0 + \sum_{i=1}^{N} \int_{s_{i-1}}^{s_i} \int_{\mathbb{X}} (y - \widehat{y}_{s_{i-1}^-})N[\widehat{\mu}_{s_{i-1}}](\mathrm{d}s, \mathrm{d}y)$$

$$= \widehat{y}_0 + \sum_{i=1}^{N} \int_{\mathbb{X}} (y - \widehat{y}_{s_{i-1}^-})N[\widehat{\mu}_{s_{i-1}}]((s_{i-1}, s_i], \mathrm{d}y),$$

which given $\mathbb{X}$ is finite, can be further decomposed into the following sum of jumps:

$$\widehat{y}_{s_N} = \widehat{y}_0 + \sum_{i=1}^{N} \int_{\mathbb{X}} (y - \widehat{y}_{s_{i-1}^-})N[\widehat{\mu}_{s_{i-1}}]((s_{i-1}, s_i], \mathrm{d}y)$$

$$= \widehat{y}_0 + \sum_{i=1}^{N} \sum_{y \in \mathbb{X}} (y - \widehat{y}_{s_{i-1}^-})N[\widehat{\mu}_{s_{i-1}}]((s_{i-1}, s_i], \{y\})$$

$$\sim \widehat{y}_0 + \sum_{i=1}^{N} \sum_{y \in \mathbb{X}} (y - \widehat{y}_{s_{i-1}^-})\mathcal{P}((s_i - s_{i-1})\widehat{\mu}_{s_{i-1}}(y)),$$

which is exactly (4.1) in the $\tau$-leaping algorithm (Algorithm 1). □

**Remark A.13.** *In Algorithm 1 and Proposition 4.1, we have implicitly assumed certain underlying additive group structure on the state space $\mathbb{X}$ such that the difference $y - \widehat{y}_{s_n}$ in (4.1) and $y - \widehat{y}_{\lfloor s \rfloor^-}$ in (4.2) and their multiples are well-defined and can be summed up with states. In certain spaces with apparent ordering, e.g. $[S]^d$, this assumption can be easily satisfied by resolving violations at the boundary of the state space with clipping operations. In practice, these discrepancies only happen with a small probability given a sufficiently small stepsize, as we will show in the following proposition.*

In order to quantify the issue in Remark A.13 and apply the change of measure argument in Theorem 3.3 to the $\tau$-leaping algorithm, we provide the following alternative stochastic integral formulation of the $\tau$-leaping algorithm.

**Proposition A.14** (Alternative Stochastic Integral Formulation of $\tau$-Leaping)**.** *Under the assumption on the time discretization scheme and the choice of parameters in Theorem 4.7, with probability $1 - \mathcal{O}(\kappa \overline{D}^2 T)$, the $\tau$-leaping algorithm (Algorithm 1) is equivalent to solving the following stochastic integral equation:*

$$\widehat{y}_s = \widehat{y}_0 + \int_0^s \int_{\mathbb{X}} (y - \widehat{y}_{\tau^-})N[\widehat{\mu}^\theta_{\lfloor \cdot \rfloor}](\mathrm{d}\tau, \mathrm{d}y), \tag{A.15}$$

*where the evolving intensity $\widehat{\mu}^\theta_\tau(y)$ is given by $\widehat{\mu}^\theta_{\lfloor \tau \rfloor}(y) = \overleftarrow{\widehat{s}}^\theta_{\lfloor \tau \rfloor}(\widehat{y}_{\tau^-}, y)\widetilde{Q}_{\lfloor \tau \rfloor}(\widehat{y}_{\tau^-}, y).$*

*Proof.* For simplicity, we first consider the stochastic integral (4.2) within the time interval $(s_n, s_{n+1}]$ and assume that $\widehat{y}$ is left-continuous at time $s_n$ which happens with probability 1.

Notice that (A.15) only differs from (4.2) in that the integrand is evaluated at each time $s$ instead of the starting time of the interval $\lfloor s \rfloor = s_n$. Consequently, $\widehat{y}_{s^-}$ coincides with $\widehat{y}_{s_n^-} = \widehat{y}_{s_n}$ before the first jump within the interval. Therefore, both the first jump time (A.13) and the first jump location (A.14) are distributed identically for the two stochastic integral formulations, and if no further jump occurs within the interval, the two formulations are equivalent within the interval.

We bound the probability of the scenario where the two formulations differ, *i.e.* at least two jumps occur within the interval $(s_n, s_{n+1}]$. We will denote this event as $A_n$. By the equivalence between Algorithm 1 and the stochastic integral (4.2), we have that

$$
\begin{aligned}
\mathbb{P}(A_n) &= \mathbb{P}\left(\sum_{y \in \mathbb{X}} \mathcal{P}\left(\widehat{\mu}_{s_n}^\theta(y)(s_{n+1} - s_n)\right) > 1\right) \\
&= 1 - \mathbb{P}\left(\mathcal{P}\left(\sum_{y \in \mathbb{X}} \widehat{\mu}_{s_n}^\theta(y)(s_{n+1} - s_n)\right) \leq 1\right) \\
&= 1 - \exp\left(-\sum_{y \in \mathbb{X}} \widehat{\mu}_{s_n}^\theta(y)(s_{n+1} - s_n)\right)\left(1 + \sum_{y \in \mathbb{X}} \widehat{\mu}_{s_n}^\theta(y)(s_{n+1} - s_n)\right) \\
&\lesssim \left(\int_{s_n}^{s_{n+1}} \int_{\mathbb{X}} \widehat{\mu}_{s_n}^\theta(y)\nu(\mathrm{d}y)\mathrm{d}s\right)^2,
\end{aligned}
$$

which, with a similar argument as in (C.1), is further bounded by

$$
\mathbb{P}(A_n) \lesssim \left(1 \vee (T - s_{n+1})^{-2}\right)\overline{D}^2(s_{n+1} - s_n)^2,
$$

and thus

$$
\begin{aligned}
\mathbb{P}\left(\bigcup_{n=1}^N A_n\right) &\lesssim \sum_{n=1}^N \left(1 \vee (T - s_{n+1})^{-2}\right)\overline{D}^2(s_{n+1} - s_n)^2 \\
&\leq \sum_{n=1}^N \left(1 \vee (T - s_{n+1})^{-1+\gamma-\eta}\right)\overline{D}^2 \kappa(s_{n+1} - s_n) \\
&\lesssim \kappa\overline{D}^2\left(T + \int_\delta^1 t^{-1+\gamma-\eta}\mathrm{d}t\right) \lesssim \kappa\overline{D}^2 T,
\end{aligned}
$$

where the last inequality is by following the same argument as in the proof of Proposition C.5 and the choice of the early stopping time $\delta$ in Theorem 4.7. The result follows. $\square$

### A.4.2 UNIFORMIZATION

**Theorem A.15** (Accurate Simulation by Uniformization)**.** *The uniformization algorithm (Algorithm 2) with its stochastic integral formulation in (4.4) is equivalent to the approximate backward process with its stochastic integral formulation in (A.12).*

*Proof of Proposition 4.2 and Theorem A.15.* For simplicity, we only consider the stochastic integral (4.4) within the time interval $(s_b, s_{b+1}]$.

We rewrite the stochastic integral (4.4) as a sum of jumps:

$$
y_{s_{b+1}} = y_{s_b} + \sum_{i=1}^N (Y_n - y_{s_{b,n}^-})\mathbf{1}_{0 \leq \Xi_n \leq \int_{\mathbb{X}} \widehat{\mu}_{s_{s_{b,n}}}^\theta(y)\nu(\mathrm{d}y)},
$$

where by Proposition A.8, $(s_{b,n})_{n \in [N]}$ are the jump times and $(Y_n, \Xi_n)_{n \in [N]}$ are the jump locations that are distributed according to

$$
\mathbb{P}(Y_n = y, \Xi_n = \xi) = \frac{\widehat{\mu}_{s_{b,n}}^\theta(y)}{\int_{\mathbb{X}} \widehat{\mu}_{s_{b,n}}^\theta(y)\nu(\mathrm{d}y)\int_0^{\overline{\lambda}}\mathrm{d}\xi} = \frac{\widehat{\mu}_{s_{b,n}}^\theta(y)}{\int_{\mathbb{X}} \widehat{\mu}_{s_{b,n}}^\theta(y)\nu(\mathrm{d}y)\overline{\lambda}}. \tag{A.16}
$$

Therefore, the $n$-th jump is not performed if $\int_{\mathbb{X}} \widehat{\mu}^{\theta}_{s_{s_{b,n}}}(y)\nu(\mathrm{d}y) < \Xi_n \leq \overline{\lambda}$, which is of probability

$$\mathbb{P}(\Xi_n > \widehat{\mu}^{\theta}_{s_{b,n}}(y_{s_{b,n}^-})) = \frac{\int_{\mathbb{X}} \widehat{\mu}^{\theta}_{s_{b,n}}(y)\nu(\mathrm{d}y)}{\int_{\mathbb{X}} \widehat{\mu}^{\theta}_{s_{b,n}}(y)\nu(\mathrm{d}y)} \frac{\overline{\lambda} - \int_{\mathbb{X}} \widehat{\mu}^{\theta}_{s_{b,n}}(y)\nu(\mathrm{d}y)}{\overline{\lambda}}$$

$$= 1 - \frac{\int_{\mathbb{X}} \widehat{\mu}^{\theta}_{s_{b,n}}(y)\nu(\mathrm{d}y)}{\overline{\lambda}},$$

and is to the state $y$ with probability

$$\mathbb{P}\left(Y_n = y, \Xi_n \leq \int_{\mathbb{X}} \widehat{\mu}^{\theta}_{s_{s_{b,n}}}(y)\nu(\mathrm{d}y)\right)$$

$$= \frac{\widehat{\mu}^{\theta}_{s_{b,n}}(y)}{\int_{\mathbb{X}} \widehat{\mu}^{\theta}_{s_{b,n}}(y)\nu(\mathrm{d}y)} \frac{\int_{\mathbb{X}} \widehat{\mu}^{\theta}_{s_{b,n}}(y)\nu(\mathrm{d}y)}{\overline{\lambda}} = \frac{\widehat{\mu}^{\theta}_{s_{b,n}}(y)}{\overline{\lambda}},$$

which coincides with (4.3) in the uniformization algorithm (Algorithm 2).

By conditioning on the occurrence of each jump, *i.e.* $0 \leq \Xi_n \leq \int_{\mathbb{X}} \widehat{\mu}^{\theta}_{s_{s_{b,n}}}(y)\nu(\mathrm{d}y)$, with slight abuse of notation, we have that

$$\mathbb{P}\left(Y_n = y \middle| 0 \leq \Xi_n \leq \int_{\mathbb{X}} \widehat{\mu}^{\theta}_{s_{s_{b,n}}}(y)\nu(\mathrm{d}y)\right) = \frac{\widehat{\mu}^{\theta}_{s_{b,n}}(y)}{\int_{\mathbb{X}} \widehat{\mu}^{\theta}_{s_{b,n}}(y)\nu(\mathrm{d}y)},$$

which again by Proposition A.8 implies that $y_s$ also satisfies the stochastic integral (A.12) corresponding to the approximate backward process, and vice versa, and the result follows. □

## B   RESULTS FOR CONTINUOUS-TIME MARKOV CHAIN

In this section, we will provide some results for the continuous-time Markov chain (CTMC), including the mixing time, the spectral gap, the modified log-Sobolev constant, *etc.*.

**Definition B.1** (Graph Corresponding to Rate Matrix). *We denote $\mathcal{G}(\boldsymbol{Q})$ as the graph corresponding to the rate matrix $\boldsymbol{Q}$,* i.e.

$$\mathcal{G}(\boldsymbol{Q}) = (\mathbb{X}, E(\mathcal{G}(\boldsymbol{Q})), Q), \ where \ E(\mathcal{G}(\boldsymbol{Q})) = \{(x,y) \in \mathbb{X} \times \mathbb{X} | x \neq y, Q(x,y) > 0\},$$

*and the weight of the directed edge $(x,y) \in E(\mathcal{G}(\boldsymbol{Q}))$ is $Q(x,y)$.*

We will assume that the continuous-time Markov chain is irreducible and reversible on the state space $\mathbb{X}$, and the corresponding stationary distribution is $\boldsymbol{\pi}$.

### B.1   SPECTRAL GAP

**Definition B.2** (Spectral Gap). *Let $\boldsymbol{L} = -\boldsymbol{Q}$ be the* graph Laplacian *matrix with $\boldsymbol{D} = \mathrm{diag}\,\boldsymbol{L}$, corresponding to the graph $\mathcal{G}(\boldsymbol{Q})$. with*

$$0 = \lambda_1(\boldsymbol{L}) < \lambda_2(\boldsymbol{L}) \leq \ldots \leq \lambda_{|\mathbb{X}|}(\boldsymbol{L}) \leq 2\max_{x \in \mathbb{X}} D(x,x) = 2\overline{D},$$

*the spectral gap $\lambda(\boldsymbol{Q})$ of the rate matrix $\boldsymbol{Q}$ is defined as the second smallest eigenvalue of the graph Laplacian $\boldsymbol{L}$, i.e. $\lambda(\boldsymbol{Q}) = \lambda_2(\boldsymbol{L})$.*

**Remark B.3** (Asymptotic Behavior of the score function $\boldsymbol{s}_t$). *Assume $\boldsymbol{Q}$ is symmetric with the following orthogonal eigendecomposition:*

$$\boldsymbol{Q} = -\boldsymbol{U}\boldsymbol{\Lambda}\boldsymbol{U}^{\top},$$

*where $\boldsymbol{U} = (\boldsymbol{u}_1, \boldsymbol{u}_2, \ldots, \boldsymbol{u}_{|\mathbb{X}|})$ is an orthogonal matrix, and the distribution $\boldsymbol{p}_t$ has the following decomposition w.r.t. the eigenvectors of the graph Laplacian $\boldsymbol{L}$:*

$$\boldsymbol{p}_t = \sum_{i=1}^{|\mathbb{X}|} \alpha_t(i)\boldsymbol{u}_i = \boldsymbol{U}\boldsymbol{\alpha}(t),$$

*then the solution to the master equation (2.7) is given by*

$$\boldsymbol{p}_t = \exp(t\boldsymbol{Q})\boldsymbol{p}_0 = \boldsymbol{U}\exp(-t\boldsymbol{\Lambda})\boldsymbol{U}^\top \boldsymbol{p}_0 = \boldsymbol{U}\exp(-t\boldsymbol{\Lambda})\boldsymbol{\alpha}_0 = \sum_{j=1}^{|\mathbb{X}|} \boldsymbol{u}_j \exp(-t\lambda_j)\boldsymbol{\alpha}_0(j),$$

*i.e.* $\boldsymbol{\alpha}_t = \exp(-t\boldsymbol{\Lambda})\boldsymbol{\alpha}_0$ *and thus for any* $i \in [|\mathbb{X}|]$,

$$p_t(i) - p_0(i) = \sum_{j=1}^{|\mathbb{X}|} u_j(i)(-1 + \exp(-t\lambda_j))\alpha_0(j) = -\sum_{j>1} u_j(i)\alpha_0(j)\lambda_j\mathcal{O}(t).$$

*Therefore, we have*

$$s_t(x,y) = \frac{p_t(y)}{p_t(x)} = \frac{p_0(y) - \sum_{j>1} u_j(y)\alpha_0(j)\lambda_j\mathcal{O}(t)}{p_0(x) - \sum_{j>1} u_j(x)\alpha_0(j)\lambda_j\mathcal{O}(t)} \lesssim 1 \vee (Ft)^{-1},$$

*given that the following condition is satisfied*

$$F = \min_{x\in\mathbb{X}} \left| \sum_{j>1} u_j(x)\alpha_0(j)\lambda_j \right| > 0,$$

*which only depends on the initial distribution* $\boldsymbol{p}_0$ *and the rate matrix* $\boldsymbol{Q}$.

*Especially, the bound* $s_t(x,y) \lesssim 1$ *for any* $x \in \mathbb{X}$ *s.t.* $p_0(x) > 0$ *and* $s_t(x,y) \lesssim t^{-1}$ *for those s.t.* $p_0(x) = 0$.

**Definition B.4** (Conductance). *The conductance* $\phi(\mathcal{G})$ *of a graph* $\mathcal{G}$ *is defined as*

$$\phi(\mathcal{G}) = \min_{S\subset\mathbb{X}} \frac{\sum_{x\in S, y\notin S} Q(x,y)}{\min\left\{\sum_{x\in S} D(x,x), \sum_{y\notin S} D(y,y)\right\}}.$$

**Theorem B.5** (Cheeger's Inequality). *Denote the* normalized graph Laplacian *matrix by* $\widetilde{\boldsymbol{L}} = \boldsymbol{D}^{-1/2}\boldsymbol{L}\boldsymbol{D}^{-1/2}$ *with eigenvalues*

$$0 \le \lambda_1(\overline{\boldsymbol{L}}) \le \lambda_2(\overline{\boldsymbol{L}}) \le \ldots \le \lambda_{|\mathbb{X}|}(\overline{\boldsymbol{L}}) \le 2,$$

*then the conductance of the graph* $\mathcal{G}(\boldsymbol{Q})$ *can be bounded by*

$$\frac{1}{2}\lambda_2(\overline{\boldsymbol{L}}) \le \phi(\mathcal{G}(\boldsymbol{Q})) \le \sqrt{2\lambda_2(\overline{\boldsymbol{L}})}.$$

### B.2 LOG-SOBOLEV INEQUALITIES

**Definition B.6** (Modified Log-Sobolev Constant (Bobkov & Tetali, 2006)). *For any function* $f, g : \mathbb{X} \to \mathbb{R}$, *we denote the* entropy functional $\mathrm{Ent}_\pi(f)$ *of* $f$ *as*

$$\mathrm{Ent}_\pi(f) := \mathbb{E}_\pi[f\log f] - \mathbb{E}_\pi[f]\log\mathbb{E}_\pi[f],$$

*and the* Dirichlet form $\mathcal{E}_\pi(f, g)$ *as*

$$\mathcal{E}_\pi(f,g) = \mathbb{E}_\pi[f\boldsymbol{L}^T g] := \sum_{y\in\mathbb{X}} f(y)(\boldsymbol{L}^T g)(y)\pi(y) = \sum_{x,y\in\mathbb{X}} f(y)L(x,y)g(x)\pi(y),$$

*where the Laplacian* $\boldsymbol{L} = \boldsymbol{Q}$. *Then the* modified log-Sobolev constant *of the rate matrix* $\boldsymbol{Q}$ *is defined as*

$$\rho(\boldsymbol{Q}) := \inf\left\{\frac{\mathcal{E}_\pi(f, \log f)}{\mathrm{Ent}_\pi(f)} \,\middle|\, f : \mathbb{X} \to \mathbb{R}, \ \mathrm{Ent}_\pi(f) > 0\right\}.$$

**Theorem B.7** (Theorem 2.4, (Bobkov & Tetali, 2006)). *For any initial distribution* $\boldsymbol{p}_0$, *we have for any* $t \ge 0$,

$$D_{\mathrm{KL}}(\boldsymbol{p}_t\|\boldsymbol{\pi}) \le D_{\mathrm{KL}}(\boldsymbol{p}_0\|\boldsymbol{\pi})\exp\left(-\rho(\boldsymbol{Q})t\right),$$

*i.e. the KL divergence converges exponentially fast with rate* $\rho(\boldsymbol{Q})$.

*Proof.* Noticing that $\text{Ent}_{\boldsymbol{\pi}}(\frac{\boldsymbol{p}_t}{\boldsymbol{\pi}}) = D_{\text{KL}}(\boldsymbol{p}_t\|\boldsymbol{\pi})$, we differentiate $D_{\text{KL}}(\boldsymbol{p}_t\|\boldsymbol{\pi})$ with respect to $t$ to obtain that

$$
\begin{aligned}
\frac{\mathrm{d}}{\mathrm{d}t} D_{\text{KL}}(\boldsymbol{p}_t\|\boldsymbol{\pi}) =& \frac{\mathrm{d}}{\mathrm{d}t} \sum_{x\in\mathbb{X}} \frac{p_t(x)}{\pi(x)} \log\left(\frac{p_t(x)}{\pi(x)}\right) \pi(x) = \sum_{x\in\mathbb{X}} \left(\log \frac{p_t(x)}{\pi(x)} + 1\right) \pi(x) \frac{\mathrm{d}}{\mathrm{d}t} \frac{p_t(x)}{\pi(x)} \\
=& \sum_{x\in\mathbb{X}} \left(\log \frac{p_t(x)}{\pi(x)} + 1\right) \frac{\mathrm{d}}{\mathrm{d}t} p_t(x) = -\sum_{x,y\in\mathbb{X}} \log \frac{p_t(x)}{\pi(x)} L(x,y) p_t(y) \\
=& -\sum_{y\in\mathbb{X}} \frac{p_t(y)}{\pi(y)} \left(\sum_{x\in\mathbb{X}} L(x,y) \log \frac{p_t(x)}{\pi(x)}\right) \pi(y) \\
=& -\mathcal{E}_{\boldsymbol{\pi}}\left(\frac{p_t}{\pi}, \log \frac{p_t}{\pi}\right) \leq -\rho(\boldsymbol{Q}) D_{\text{KL}}(\boldsymbol{p}_t\|\boldsymbol{\pi}),
\end{aligned}
\tag{B.1}
$$

and the result follows by applying Grönwall's inequality to both sides above. $\qquad\square$

Then, the following proposition connects the modified log-Sobolev constant with the spectral gap.

**Proposition B.8** ((Bobkov & Tetali, 2006, Proposition 3.5)). *The modified log-Sobolev constant $\rho(\boldsymbol{Q})$ of the rate matrix $\boldsymbol{Q}$ is bounded by the spectral gap $\lambda(\boldsymbol{Q})$, i.e. $\rho(\boldsymbol{Q}) \leq \lambda(\boldsymbol{Q})$.*

*Proof.* Below we provide a sketch of the informal proof of the proposition above for the sake of completeness. Let $f : \mathbb{X} \to \mathbb{R}$ be an arbitrary function and $\zeta > 0$ be any positive number. From the definition of the modified log-Sobolev constant, we have

$$
\mathcal{E}_{\boldsymbol{\pi}}(e^{\zeta f}, \zeta f) \geq \rho(\boldsymbol{Q}) \text{Ent}_{\boldsymbol{\pi}}(e^{\zeta f}).
\tag{B.2}
$$

Under the limit $\zeta \to 0^+$, we may apply Taylor expansion to the two terms on the LHS and RHS, which implies

$$
\begin{aligned}
\mathcal{E}_{\boldsymbol{\pi}}(e^{\zeta f}, \zeta f) &= \mathcal{E}_{\boldsymbol{\pi}}(1 + \zeta f + O(\zeta^2), \zeta f) \\
&= \zeta \mathcal{E}_{\boldsymbol{\pi}}(1, f) + \zeta^2 \mathcal{E}_{\boldsymbol{\pi}}(f, f) + O(\zeta^3) = \zeta^2 \mathcal{E}_{\boldsymbol{\pi}}(f, f) + O(\zeta^3) \\
\text{Ent}_{\boldsymbol{\pi}}(e^{\zeta f}) &= \mathbb{E}_{\boldsymbol{\pi}}[e^{\zeta f} \zeta f] - \mathbb{E}_{\boldsymbol{\pi}}[e^{\zeta f}] \log \mathbb{E}_{\boldsymbol{\pi}}[e^{\zeta f}] \\
&= \mathbb{E}_{\boldsymbol{\pi}}\left[(1 + \zeta f + O(\zeta^2)) \zeta f\right] - \mathbb{E}_{\boldsymbol{\pi}}[1 + \zeta f + O(\zeta^2)] \log \mathbb{E}_{\boldsymbol{\pi}}[e^{\zeta f}] \\
&= \zeta \mathbb{E}_{\boldsymbol{\pi}}[f] + \zeta^2 \mathbb{E}_{\boldsymbol{\pi}}[f^2] + O(\zeta^3) \\
&\quad - (1 + \zeta \mathbb{E}_{\boldsymbol{\pi}}[f] + O(\zeta^2)) \log (1 + \zeta \mathbb{E}_{\boldsymbol{\pi}}[f] + O(\zeta^2)) \\
&= \zeta \mathbb{E}_{\boldsymbol{\pi}}[f] + \zeta^2 \mathbb{E}_{\boldsymbol{\pi}}[f^2] + O(\zeta^3) - (1 + \zeta \mathbb{E}_{\boldsymbol{\pi}}[f] + O(\zeta^2)) (\zeta \mathbb{E}_{\boldsymbol{\pi}}[f] + O(\zeta^2)) \\
&= \zeta^2 (\mathbb{E}_{\boldsymbol{\pi}}[f^2] - \mathbb{E}_{\boldsymbol{\pi}}[f]^2) + O(\zeta^3)
\end{aligned}
\tag{B.3}
$$

Substituting (B.3) into (B.2) and taking the limit $\zeta \to 0^+$ then yield the following inequality

$$
\rho(Q) \leq \frac{\mathcal{E}_{\boldsymbol{\pi}}(f, f)}{\mathbb{E}_{\boldsymbol{\pi}}[f^2] - \mathbb{E}_{\boldsymbol{\pi}}[f]^2}
$$

for any non-constant function $f : \mathbb{X} \to \mathbb{R}$. Taking infimum on both sides above with respect to all $f$ then indicates

$$
\rho(Q) \leq \inf_f \frac{\mathcal{E}_{\boldsymbol{\pi}}(f, f)}{\mathbb{E}_{\boldsymbol{\pi}}[f^2] - \mathbb{E}_{\boldsymbol{\pi}}[f]^2} \leq \inf_{f:\mathbb{E}_{\boldsymbol{\pi}}[f]=0} \frac{\mathcal{E}_{\boldsymbol{\pi}}(f, f)}{\mathbb{E}_{\boldsymbol{\pi}}[f^2]} = \lambda_2(\boldsymbol{L}) = \lambda(\boldsymbol{Q})
\tag{B.4}
$$

where the last two equalities above follows from the definition of spectral gap, as desired. $\qquad\square$

In general, the lower bound of the modified log-Sobolev constant $\rho(\boldsymbol{Q})$ and the spectral gap $\lambda(\boldsymbol{Q})$ depends on the connectivity and other specific structures of the graph $\mathcal{G}(\boldsymbol{Q})$, and the related research is still an active area on a graph-by-graph basis (Bobkov & Tetali, 2006).

The properties of the spectral gap $\lambda(\boldsymbol{Q})$ are better known in the literature, as it is closely related to the conductance of the graph $\mathcal{G}(\boldsymbol{Q})$ via Cheeger's inequality (Theorem B.5), and thus when $\boldsymbol{D} = D\boldsymbol{I}$, the spectral gap $\lambda(\boldsymbol{Q})$ satisfies

$$
\frac{1}{2D} \lambda(\boldsymbol{Q}) = \frac{1}{2} \lambda_2(\overline{\boldsymbol{L}}) \leq \phi(\mathcal{G}(\boldsymbol{Q})) \leq \sqrt{2\lambda_2(\overline{\boldsymbol{L}})} = \sqrt{\frac{2\lambda(\boldsymbol{Q})}{D}}.
$$

However, as shown in Proposition B.8, the lower bound on the modified log-Sobolev constant $\rho(\boldsymbol{Q})$ is hard to obtain, as the KL divergence, the exponential convergence of which is controlled by $\rho(\boldsymbol{Q})$, is stronger than the total variation distance, the exponential convergence of which is controlled by $\lambda(\boldsymbol{Q})$, via Pinsker's inequality. The following theorem gives a rough lower bound on $d$-regular graphs.

**Theorem B.9** ((Bobkov & Tetali, 2006, Proposition 5.4)). *Suppose $\mathcal{G}$ is a $d$-regular graph on $\mathbb{X}$ with unit weights and $\boldsymbol{Q}$ is the corresponding rate matrix such that $\mathcal{G}(\boldsymbol{Q}) = \mathcal{G}$, then the modified log-Sobolev constant $\rho(\boldsymbol{Q})$ of the rate matrix $\boldsymbol{Q}$ satisfies*

$$\frac{\lambda(\boldsymbol{Q})}{\log |\mathbb{X}|} \le \rho(\boldsymbol{Q}) \le \frac{8d \log |\mathbb{X}|}{\mathrm{diam}(\mathcal{G})^2},$$

*where $\mathrm{diam}(\mathcal{G})$ is the diameter of the graph $\mathcal{G}$.*

For some specific graphs, the modified log-Sobolev constant $\rho(\boldsymbol{Q})$ and the spectral gap $\lambda(\boldsymbol{Q})$ can be explicitly calculated, such as the following examples:

**Example B.10** (Hypercube (Gross, 1975)). *Let $\mathbb{X} = \{0,1\}^d$ and $\boldsymbol{Q}$ be the rate matrix for which the graph $\mathcal{G}(\boldsymbol{Q})$ is a hypercube, and for any two states $x, y \in \mathbb{X}$, the rate $Q(x, y) = 1$ if $x$ and $y$ differ in exactly one coordinate. Then the modified log-Sobolev constant $\rho(\boldsymbol{Q})$ and the spectral gap $\lambda(\boldsymbol{Q})$ are given by*

$$\rho(\boldsymbol{Q}) = \lambda(\boldsymbol{Q}) = 4,$$

*which is dimensionless.*

**Example B.11** (Asymmetric Hypercube (Diaconis & Saloff-Coste, 1996)). *Let $\mathbb{X} = \{0,1\}^d$ and $\boldsymbol{Q}$ be the rate matrix for which the graph $\mathcal{G}(\boldsymbol{Q})$ is a hypercube, and for any two states $x, y \in \mathbb{X}$, the rate $Q(x, y) = p$ if $x$ and $y$ differ in exactly one coordinate and $x$ is the state with $0$ in that coordinate, and $Q(x, y) = q = 1 - p$ if with $1$ in that coordinate. Then the modified log-Sobolev constant $\rho(\boldsymbol{Q})$ and the spectral gap $\lambda(\boldsymbol{Q})$ are given by*

$$\rho(\boldsymbol{Q}) = \frac{2(p - q)}{pq(\log p - \log q)}, \text{ and } \lambda(\boldsymbol{Q}) = \frac{1}{pq}.$$

Further results on log-Sobolev inequalities related to finite-state Markov chains are beyond the scope of this paper, and we refer the readers to (Stroock, 1993; Saloff-Coste, 1997; Bobkov & Ledoux, 1998; Lee & Yau, 1998; Goel, 2004; Ledoux, 2006) for more detail.

## B.3 MIXING TIME

**Definition B.12** (Mixing Time). *We define the mixing time $t_{\mathrm{mix}}(\epsilon)$ of the continuous-time Markov chain with rate matrix $\boldsymbol{Q}$ as the smallest time $t$ such that starting from any initial distribution $\boldsymbol{p}_0$, the KL divergence $D_{\mathrm{KL}}(\boldsymbol{p}_t \| \boldsymbol{\pi})$ is less than $\epsilon$, i.e.*

$$t_{\mathrm{mix}}(\epsilon) = \inf \left\{ t \in \mathbb{R}_+ \,\middle|\, D_{\mathrm{KL}}(\boldsymbol{p}_t \| \boldsymbol{\pi}) = D_{\mathrm{KL}}(e^{-t\boldsymbol{Q}} \boldsymbol{p}_0 \| \boldsymbol{\pi}) \le \epsilon \right\}.$$

*Similarly, we define the mixing time $t_{\mathrm{mix,TV}}(\epsilon)$ as the smallest time $t$ such that starting from any initial distribution $\boldsymbol{p}_0$, the total variation distance $\mathrm{TV}(\boldsymbol{p}_t, \boldsymbol{\pi})$ is less than $\epsilon$, i.e.*

$$t_{\mathrm{mix,TV}}(\epsilon) = \inf \left\{ t \in \mathbb{R}_+ \,\middle|\, \mathrm{TV}(\boldsymbol{p}_t, \boldsymbol{\pi}) = \mathrm{TV}(e^{-t\boldsymbol{Q}} \boldsymbol{p}_0, \boldsymbol{\pi}) \le \epsilon \right\}.$$

*With a slight abuse of notation, we will also denote the $e^{-1}$-mixing time as $t_{\mathrm{mix}} = t_{\mathrm{mix,KL}}(e^{-1})$ and $t_{\mathrm{mix,TV}} = t_{\mathrm{mix,TV}}(e^{-1})$.*

**Proposition B.13.** *The mixing time $t_{\mathrm{mix}}(\epsilon)$ of the continuous-time Markov chain with rate matrix $\boldsymbol{Q}$ is bounded by the modified log-Sobolev constant $\rho(\boldsymbol{Q})$, i.e.*

$$t_{\mathrm{mix}}(\epsilon) \lesssim \rho(\boldsymbol{Q})^{-1} \left( \log \epsilon^{-1} + \log \log \pi_*^{-1} \right),$$

*And the mixing time $t_{\mathrm{mix,TV}}(\epsilon)$ is bounded by the spectral gap $\lambda(\boldsymbol{Q})$, i.e.*

$$t_{\mathrm{mix,TV}}(\epsilon) \lesssim \lambda(\boldsymbol{Q})^{-1} \left( \log \epsilon^{-1} + \log \log \pi_*^{-1} \right).$$

*Proof.* Define $\pi_* = \min_{x \in \mathbb{X}} \pi(x)$, we first bound $D_{\mathrm{KL}}(\boldsymbol{p}_0 \| \boldsymbol{\pi})$ as follows:

$$D_{\mathrm{KL}}(\boldsymbol{p}_0 \| \boldsymbol{p}_\infty) \leq \sum_{x \in \mathbb{X}} p_0(x) \log \frac{p_0(x)}{\pi(x)} \leq \log \pi_*^{-1}, \tag{B.5}$$

and thus by Theorem B.7, we have

$$D_{\mathrm{KL}}(\boldsymbol{p}_t \| \boldsymbol{\pi}) \leq D_{\mathrm{KL}}(\boldsymbol{p}_0 \| \boldsymbol{\pi}) \exp\left(-\rho(\boldsymbol{Q})t\right) \leq \log \pi_*^{-1} \exp\left(-\rho(\boldsymbol{Q})t\right). \tag{B.6}$$

Therefore, by setting the right-hand side of (B.6) to be $\epsilon$, we have the desired result for the mixing time

$$t_{\mathrm{mix}}(\epsilon) \leq \frac{1}{\rho(\boldsymbol{Q})} \left(\log \epsilon^{-1} + \log \log \pi_*^{-1}\right).$$

For the mixing time $t_{\mathrm{mix,TV}}(\epsilon)$, we use the Pinsker's inequality to obtain:

$$\mathrm{TV}(\boldsymbol{p}_t, \boldsymbol{\pi}) \leq \sqrt{\frac{1}{2} D_{\mathrm{KL}}(\boldsymbol{p}_t \| \boldsymbol{\pi})} \leq \sqrt{\frac{1}{2} \log \pi_*^{-1}} \exp\left(-\rho(\boldsymbol{Q})t\right),$$

and therefore,

$$t_{\mathrm{mix,TV}}(\epsilon) \lesssim \frac{1}{\rho(\boldsymbol{Q})} \log\left(\epsilon^{-1} \sqrt{\log \pi_*^{-1}}\right) \lesssim \frac{1}{\rho(\boldsymbol{Q})} \left(\log \epsilon^{-1} + \log \log \pi_*^{-1}\right).$$

$\square$

**Corollary B.14.** *The $e^{-1}$-mixing time $t_{\mathrm{mix}}$ and $t_{\mathrm{mix,TV}}$ of the continuous-time Markov chain with rate matrix $\boldsymbol{Q}$ satisfy*

$$t_{\mathrm{mix}} \lesssim \rho(\boldsymbol{Q})^{-1} \log \log \pi_*^{-1}, \quad \text{and} \quad t_{\mathrm{mix,TV}} \lesssim \rho(\boldsymbol{Q})^{-1} \log \log \pi_*^{-1},$$

*and thus $t_{\mathrm{mix}} \gtrsim t_{\mathrm{mix,TV}}$.*

## C    PROOFS OF ERROR ANALYSIS IN SECTION 4.3

In this section, we provide the proof of the primary results in the main text and their sketches.

### C.1    PROOF SKETCH

A general pipeline of the proofs of Theorem 4.7 and 4.9 is to decompose the KL divergence between the target distribution and the distribution of generated sample as a summation of the truncation, approximation, and discretization errors, and then bound each term separately, echoing that for the continuous case in Section 2.3.

- **Truncation Error:** As in the continuous case, the truncation error is caused by the finite time horizon of the forward process (2.7) and is thus bounded by the convergence rate of the constructed forward process. In our work, we use the modified log-Sobolev constant $\rho(\boldsymbol{Q})$ to bound the truncation error (*cf.* Definition B.6). As discussed in Appendix B, a lower bound on the modified log-Sobolev constant $\rho(\boldsymbol{Q})$ guarantees the exponential convergence of the forward process (*cf.* Theorem B.7), which then implies the upper bound on the truncation error presented in Proposition C.1.

- **Discretization Error:** In the $\tau$-leaping algorithm, the discretization error is caused by the approximation of the backward process with a piecewise constant function, and should decrease as the time step $\kappa$ decreases. We first analyze the discretization error within one step (*cf.* Proposition C.4) and then provide the overall discretization error in Proposition C.5, taking different discretization scheme into consideration and deriving early stopping criteria. In the uniformization algorithm, due to its exactness, the discretization error is zero.

- **Approximation Error:** The approximation error is caused by the estimation error of the score function, and is bounded by certain error assumptions (*cf.* Assumption 4.6 for $\tau$-leaping and Assumption 4.6' for uniformization) on the score function estimation in the training process.

Then in Appendix C.4, we first invoke the change of measure theorem 3.3 and obtain KL divergence bounds for both algorithms (Theorem C.6 and C.7), and then provide the overall error bounds thereafter.

## C.2 TRUNCATION ERROR

**Theorem C.1** (Truncation Error). *The forward process (2.7) converges to the uniform distribution* $\boldsymbol{p}_\infty = \mathbf{1}/|\mathbb{X}|$ *exponentially fast in terms of the KL divergence,* i.e.

$$D_{\mathrm{KL}}(\boldsymbol{p}_t \| \boldsymbol{p}_\infty) = D_{\mathrm{KL}}\left(\boldsymbol{p}_t \Big\| \frac{\mathbf{1}}{|\mathbb{X}|}\right) \lesssim e^{-\rho t} \log |\mathbb{X}|,$$

*where* $|\mathbb{X}|$ *is the size of the state space, and* $t_{\mathrm{mix}}$ *is the mixing time of the continuous-time Markov chain corresponding to the rate matrix* $\boldsymbol{Q}$ *defined in Definition B.12.*

*Proof.* Since $\boldsymbol{Q}$ is symmetric, we have the stationary distribution $\boldsymbol{\pi} = \mathbf{1}/|\mathbb{X}|$ and thus $D_{\mathrm{KL}}(\boldsymbol{p}_t \| \boldsymbol{p}_\infty) = D_{\mathrm{KL}}(\boldsymbol{p}_t \| \boldsymbol{\pi})$, and $\pi_* = 1/|\mathbb{X}|$. By Assumption 4.3 and Theorem B.7, we have

$$D_{\mathrm{KL}}(\boldsymbol{p}_t \| \boldsymbol{\pi}) \leq e^{-\rho(\boldsymbol{Q})t} D_{\mathrm{KL}}(\boldsymbol{p}_0 \| \boldsymbol{\pi}) \leq e^{-\rho t} \log \pi_*^{-1} \leq e^{-\rho t} \log |\mathbb{X}|,$$

where the last inequality is by (B.5). $\qquad\square$

## C.3 DISCRETIZATION ERROR

Denote the shorthand notation $G(x; y) = x(\log x - \log y) - x$; it is easy to check that $G'(x; y) = \log x - \log y$.

**Proposition C.2.** *For any* $y \in \mathbb{X}$*, we have*

$$|\partial_\sigma G(\mu_\sigma(y); \widehat{\mu}^\theta_{\lfloor s_n \rfloor}(y))| \lesssim (\log C + \log(1 \vee (T - \sigma)) + \log M) \, \mu_\sigma(y) \overline{D} \left(1 \vee (T - \sigma)^{-2}\right).$$

*Proof.* By the chain rule, we have

$$\partial_\sigma G(\mu_\sigma(y); \widehat{\mu}^\theta_{\lfloor s_n \rfloor}(y)) = G'(\mu_\sigma(y); \widehat{\mu}^\theta_{\lfloor s_n \rfloor}(y)) \partial_\sigma \mu_\sigma(y) = \left(\log \mu_\sigma(y) - \log \widehat{\mu}^\theta_{\lfloor s_n \rfloor}(y)\right) \partial_\sigma \mu_\sigma(y).$$

We first compute $\partial_\sigma \mu_\sigma(y)$ as

$$\partial_\sigma \mu_\sigma(y) = \widetilde{Q}(\breve{x}_{\sigma^-}, y) \partial_\sigma \breve{s}_\sigma(\breve{x}_{\sigma^-}, y) = \widetilde{Q}(\breve{x}_{\sigma^-}, y) \partial_\sigma \left(\frac{\breve{p}_\sigma(y)}{\breve{p}_\sigma(x_{\sigma^-})}\right)$$

$$= \widetilde{Q}(\breve{x}_{\sigma^-}, y) \left(\frac{1}{\breve{p}_\sigma(x_{\sigma^-})} \partial_\sigma \breve{p}_\sigma(y) - \frac{\breve{p}_\sigma(y)}{\breve{p}_\sigma(x_{\sigma^-})^2} \partial_\sigma \breve{p}_\sigma(x_{\sigma^-})\right)$$

$$= \widetilde{Q}(\breve{x}_{\sigma^-}, y) \left(-\frac{\breve{p}_\sigma(y)}{\breve{p}_\sigma(x_{\sigma^-})} \sum_{y' \in \mathbb{X}} \frac{\breve{p}_\sigma(y')}{\breve{p}_\sigma(y)} Q(y, y') + \frac{\breve{p}_\sigma(y)}{\breve{p}_\sigma(x_{\sigma^-})} \sum_{y' \in \mathbb{X}} \frac{\breve{p}_\sigma(y')}{\breve{p}_\sigma(x_{\sigma^-})} Q(x_{\sigma^-}, y')\right)$$

$$= \mu_\sigma(y) \left(-\sum_{y' \in \mathbb{X}} \breve{s}_\sigma(y, y') Q(y, y') + \sum_{y' \in \mathbb{X}} \breve{s}_\sigma(x_{\sigma^-}, y') Q(x_{\sigma^-}, y')\right),$$

by which we have

$$|\partial_\sigma \mu_\sigma(y)| \lesssim \mu_\sigma(y) \left(\sum_{y' \in \mathbb{X}} \left(1 \vee (T - \sigma)^{-2}\right) |Q(x_{\sigma^-}, y')|\right) \lesssim \mu_\sigma(y) \overline{D} \left(1 \vee (T - \sigma)^{-2}\right),$$

and thus

$$\left|\partial_\sigma G(\mu_\sigma(y); \widehat{\mu}^\theta_{\lfloor s_n \rfloor}(y))\right| \leq \left|\log \mu_\sigma(y) - \log \widehat{\mu}^\theta_{\lfloor s_n \rfloor}(y)\right| |\partial_\sigma \mu_\sigma(y)|$$

$$\leq \left(|\log \widetilde{Q}(\breve{x}_{\sigma^-}, y)| + |\log \breve{s}_\sigma(\breve{x}_{\sigma^-}, y)| + |\log \widetilde{Q}(\breve{x}_{s_n^-}, y)| + |\log \breve{s}^\theta_{s_n}(\breve{x}_{s_n^-}, y)|\right) |\partial_\sigma \mu_\sigma(y)|$$

$$\lesssim \mu_\sigma(y) \left(\log C + \log\left(1 \vee (T - \sigma)^{-1}\right) + \log M\right) \overline{D} \left(1 \vee (T - \sigma)^{-2}\right).$$

$\qquad\square$

**Proposition C.3.** *For any $0 < s < t \leq T$, we have*

$$\int_s^t \int_{\mathbb{X}} \mu_\sigma(y') \nu(\mathrm{d}y') \mathrm{d}\sigma \lesssim \left(1 \vee (T-t)^{-1}\right) \overline{D}(t-s).$$

*Proof.*

$$\int_s^t \int_{\mathbb{X}} \mu_\sigma(y') \nu(\mathrm{d}y') \mathrm{d}\sigma = \int_s^t \int_{\mathbb{X}} \overleftarrow{s}_\sigma(\overleftarrow{x}_{\sigma-}, y') \widetilde{Q}(\overleftarrow{x}_{\sigma-}, y') \nu(\mathrm{d}y') \mathrm{d}\sigma$$

$$\lesssim \left(1 \vee (T-t)^{-1}\right) \int_s^t \int_{\mathbb{X}} \widetilde{Q}(y, \overleftarrow{x}_{\sigma-}) \nu(\mathrm{d}y') \mathrm{d}\sigma \tag{C.1}$$

$$\lesssim \left(1 \vee (T-t)^{-1}\right) \int_s^t |Q(\overleftarrow{x}_{\sigma-}, \overleftarrow{x}_{\sigma-})| \, \mathrm{d}\sigma \lesssim \left(1 \vee (T-t)^{-1}\right) \overline{D}(t-s).$$

$\square$

**Proposition C.4.** *For any $y \in \mathbb{X}$, we have*

$$\mathbb{E}\left[\left|G(\mu_s; \widehat{\mu}^\theta_{\lfloor s_n \rfloor}(y)) - G(\mu_{s_n}; \widehat{\mu}^\theta_{\lfloor s \rfloor}(y))\right|\right]$$
$$\lesssim \left(\log C + \log\left(1 \vee (T - s_{n+1})^{-1}\right) + \log M\right) \mu_\sigma(y) \overline{D}(s - s_n) \left(1 \vee (T - s_{n+1})^{-1-\gamma}\right).$$

*Proof.* Applying Theorem A.10 to the backward process (3.2), we have

$$G(\mu_{s_n}(y); \widehat{\mu}^\theta_{\lfloor s_n \rfloor}(y)) = G(\mu_{s_n}(y); \widehat{\mu}^\theta_{\lfloor s \rfloor}(y)) + \int_{s_n}^s \partial_\sigma G(\mu_\sigma(y); \widehat{\mu}^\theta_{\lfloor s_n \rfloor}(y)) \mathrm{d}\sigma$$
$$+ \int_{s_n}^s \int_{\mathbb{X}} \left(G(\mu_{\sigma+}(y); \widehat{\mu}^\theta_{\lfloor s_n \rfloor}(y)) - G(\mu_\sigma(y); \widehat{\mu}^\theta_{\lfloor s_n \rfloor}(y))\right) N[\mu](\mathrm{d}\sigma, \mathrm{d}y'),$$

where we adopt the notation $\mu_{\sigma+}(y)$ as the right limit of the càglàd process $\mu_\sigma(y)$, *i.e.* $\mu_{\sigma+}(y) = \overleftarrow{s}_\sigma(\overleftarrow{x}_\sigma, y) \widetilde{Q}(\overleftarrow{x}_\sigma, y)$.

For the first term, we have by Proposition C.2 that

$$\left|\int_{s_n}^s \partial_\sigma G(\mu_\sigma(y); \widehat{\mu}^\theta_{\lfloor s_n \rfloor}(y)) \mathrm{d}\sigma\right| \lesssim \int_{s_n}^s \left|\partial_\sigma G(\mu_\sigma(y); \widehat{\mu}^\theta_{\lfloor s_n \rfloor}(y))\right| \mathrm{d}\sigma$$
$$\lesssim \int_{s_n}^s \left(\log C + \log\left(1 \vee (T - \sigma)^{-1}\right) + \log M\right) \mu_\sigma(y) \overline{D}\left(1 \vee (T - \sigma)^{-2}\right) \mathrm{d}\sigma$$
$$\lesssim \left(\log C + \log\left(1 \vee (T - s_{n+1})^{-1}\right) + \log M\right) \mu_\sigma(y) \overline{D}(s - s_n) \left(1 \vee (T - s_{n+1})^{-1}\right).$$

For the second term, we have

$$\left|G(\mu_{\sigma+}(y); \widehat{\mu}^\theta_{\lfloor s_n \rfloor}(y)) - G(\mu_\sigma(y); \widehat{\mu}^\theta_{\lfloor s_n \rfloor}(y))\right|$$
$$= \left|G'(\xi; \widehat{\mu}^\theta_{\lfloor s_n \rfloor}(y))(\mu_{\sigma+}(y) - \mu_\sigma(y))\right| = \left|(\log \xi - \log \widehat{\mu}^\theta_{\lfloor s_n \rfloor}(y))(\mu_{\sigma+}(y) - \mu_\sigma(y))\right|$$
$$\leq \left(|\log \mu_{\sigma+}(y)| + |\log \mu_\sigma(y)| + \left|\log \widehat{\mu}^\theta_{\lfloor s_n \rfloor}(y)\right|\right) \left|\frac{\mu_{\sigma+}(y)}{\mu_\sigma(y)} - 1\right| \mu_\sigma(y)$$
$$\lesssim \left(\log C + \log\left(1 \vee (T - \sigma)^{-1}\right) + \log M\right) \left(1 \vee (T - \sigma)^{-\gamma}\right) \mu_\sigma(y),$$

where the first equality follows from the mean value theorem and the last inequality is by Assumption 4.5.

Therefore,

$$\mathbb{E}\left[\left|G(\mu_s;\widehat{\mu}^\theta_{\lfloor s_n\rfloor}(y)) - G(\mu_{s_n};\widehat{\mu}^\theta_{\lfloor s\rfloor}(y))\right|\right]$$

$$\leq\mathbb{E}\left[\left|\int_{s_n}^s \partial_\sigma G(\mu_\sigma(y);\widehat{\mu}^\theta_{\lfloor s_n\rfloor}(y))\mathrm{d}\sigma\right|\right]$$

$$+\mathbb{E}\left[\int_{s_n}^s\int_\mathbb{X}\left|G(\mu_{\sigma^+}(y);\widehat{\mu}^\theta_{\lfloor s_n\rfloor}(y)) - G(\mu_\sigma(y);\widehat{\mu}^\theta_{\lfloor s_n\rfloor}(y))\right|N[\mu](\mathrm{d}\sigma,\mathrm{d}y')\right]$$

$$\lesssim\left(\log C + \log\left(1\vee(T-s_{n+1})^{-1}\right) + \log M\right)\mu_\sigma(y)\overline{D}(s-s_n)\left(1\vee(T-s_{n+1})^{-1}\right)$$

$$+\int_{s_n}^s\int_\mathbb{X}\left(\log C + \log\left(1\vee(T-\sigma)^{-1}\right) + \log M\right)\mu_\sigma(y)\left(1\vee(T-\sigma)^{-\gamma}\right)\mu_\sigma(y')\nu(\mathrm{d}y')\mathrm{d}\sigma$$

$$\lesssim\left(\log C + \log\left(1\vee(T-s_{n+1})^{-1}\right) + \log M\right)\mu_\sigma(y)\overline{D}(s-s_n)\left(1\vee(T-s_{n+1})^{-1}\right)$$

$$+\left(\log C + \log\left(1\vee(T-s_{n+1})^{-1}\right) + \log M\right)\mu_\sigma(y)\left(1\vee(T-s_{n+1})^{-1-\gamma}\right)(s-s_n)\overline{D}$$

$$\lesssim\left(\log C + \log\left(1\vee(T-s_{n+1})^{-1}\right) + \log M\right)\mu_\sigma(y)\overline{D}(s-s_n)\left(1\vee(T-s_{n+1})^{-1-\gamma}\right),$$

where the second to last inequality is by Proposition C.3. $\qquad\square$

**Proposition C.5** (Discretization Error). *The following bound holds*

$$\int_0^{T-\delta}\int_\mathbb{X}\left|G(\mu_s(y);\widehat{\mu}^\theta_{\lfloor s_n\rfloor}(y)) - G(\mu_{s_n}(y);\widehat{\mu}^\theta_{\lfloor s\rfloor}(y))\right|\nu(\mathrm{d}y)\mathrm{d}s$$

$$\lesssim\begin{cases}\overline{D}^2\kappa T, & \gamma < 1,\\ \overline{D}^2\kappa\left(T + \log^2\delta^{-1}\right), & \gamma = 1,\end{cases}$$

*with* $N = \begin{cases}\kappa^{-1}T, & \gamma < 1\\ \kappa^{-1}(T + \log\delta^{-1}), & \gamma = 1\end{cases}$ *steps, by taking* $\gamma < \eta \lesssim 1 - T^{-1}$ *when* $\gamma < 1$, *and* $\eta = 1$ *when* $\gamma = 1$. *In particular, in the former case, early stopping at time* $T - \delta$ *is not necessary,* i.e. $\delta = 0$.

*Proof.* We have by Proposition C.4 that

$$\int_{s_n}^{s_{n+1}}\int_\mathbb{X}\left|G(\mu_s(y);\widehat{\mu}^\theta_{\lfloor s_n\rfloor}(y)) - G(\mu_{s_n}(y);\widehat{\mu}^\theta_{\lfloor s\rfloor}(y))\right|\nu(\mathrm{d}y)\mathrm{d}s$$

$$\lesssim\int_{s_n}^{s_{n+1}}\int_\mathbb{X}\mu_\sigma(y)\nu(\mathrm{d}y)$$

$$\left(\log C + \log\left(1\vee(T-s_{n+1})^{-1}\right) + \log M\right)\overline{D}(s-s_n)\left(1\vee(T-s_{n+1})^{-1-\gamma}\right)\mathrm{d}s$$

$$\lesssim\left(\log C + \log\left(1\vee(T-s_{n+1})^{-1}\right) + \log M\right)\overline{D}^2(s_{n+1}-s_n)^2\left(1\vee(T-s_{n+1})^{-1-\gamma}\right).$$

- *Case 1:* $\underline{\gamma < \eta \lesssim 1 - T^{-1}}$

  The following bound holds

$$\int_{s_n}^{s_{n+1}}\int_\mathbb{X}\left|G(\mu_s(y);\widehat{\mu}^\theta_{\lfloor s_n\rfloor}(y)) - G(\mu_{s_n}(y);\widehat{\mu}^\theta_{\lfloor s\rfloor}(y))\right|\nu(\mathrm{d}y)\mathrm{d}s$$

$$\lesssim\left(1 + \log\left(1\vee(T-s_{n+1})^{-1}\right)\right)\overline{D}^2\kappa(s_{n+1}-s_n)\left(1\vee(T-s_{n+1})^{-1-\gamma+\eta}\right),$$

and thus, the following error

$$\sum_{n=0}^{N-1} \int_{s_n}^{s_{n+1}} \int_{\mathbb{X}} \left| G(\mu_s(y); \widehat{\mu}^\theta_{\lfloor s_n \rfloor}(y)) - G(\mu_{s_n}(y); \widehat{\mu}^\theta_{\lfloor s \rfloor}(y)) \right| \nu(\mathrm{d}y)\mathrm{d}s$$

$$\lesssim \sum_{n=0}^{N-1} \left(1 + \log\left(1 \vee (T - s_{n+1})^{-1}\right)\right) \overline{D}^2 \kappa(s_{n+1} - s_n) \left(1 \vee (T - s_{n+1})^{-1-\gamma+\eta}\right)$$

$$\lesssim \overline{D}^2 \kappa \left(T + \int_\delta^1 t^{-1-\gamma+\eta} \log t^{-1} \mathrm{d}t\right) \lesssim \overline{D}^2 \kappa \left(T + \int_1^{\delta^{-1}} t^{-1-(\eta-\gamma)} \log t \mathrm{d}t\right)$$

$$\lesssim \overline{D}^2 \kappa \left(T + \delta^{\eta-\gamma} \log \delta^{-1}\right) \to \overline{D}^2 \kappa T, \quad \text{as } \delta \to 0,$$

is achievable with finite number of steps $N$, *i.e.*

$$N \lesssim \int_\delta^T \frac{1}{\kappa(1 \wedge t^\eta)} \mathrm{d}t \lesssim \kappa^{-1}T + \kappa^{-1} \int_\delta^1 t^{-\eta} \mathrm{d}t \lesssim \kappa^{-1}\left(T + \frac{1}{1-\eta}\right) \lesssim \kappa^{-1}T,$$

where we take $\eta \lesssim 1 - T^{-1}$.

- *Case 2: $\gamma = \eta = 1$*

  We have the following bound

  $$\int_{s_n}^{s_{n+1}} \int_{\mathbb{X}} \left| G(\mu_s(y); \widehat{\mu}^\theta_{\lfloor s_n \rfloor}(y)) - G(\mu_{s_n}(y); \widehat{\mu}^\theta_{\lfloor s \rfloor}(y)) \right| \nu(\mathrm{d}y)\mathrm{d}s$$

  $$\lesssim \left(1 + \log\left(1 \vee (T - s_{n+1})^{-1}\right)\right) \overline{D}^2 \kappa(s_{n+1} - s_n) \left(1 \vee (T - s_{n+1})^{-1}\right),$$

  and similarly

  $$\sum_{n=0}^{N-1} \int_{s_n}^{s_{n+1}} \int_{\mathbb{X}} \left| G(\mu_s(y); \widehat{\mu}^\theta_{\lfloor s_n \rfloor}(y)) - G(\mu_{s_n}(y); \widehat{\mu}^\theta_{\lfloor s \rfloor}(y)) \right| \nu(\mathrm{d}y)\mathrm{d}s$$

  $$\lesssim \sum_{n=0}^{N-1} \left(1 + \log\left(1 \vee (T - s_{n+1})^{-1}\right)\right) \overline{D}^2 \kappa(s_{n+1} - s_n) \left(1 \vee (T - s_{n+1})^{-1}\right)$$

  $$\lesssim \overline{D}^2 \kappa \left(T + \int_1^{\delta^{-1}} t^{-1} \log t \mathrm{d}t\right) \lesssim \overline{D}^2 \kappa \left(T + \log^2 \delta^{-1}\right).$$

  However, the number of steps $N$ is now bounded by

  $$N \lesssim \int_\delta^T \frac{1}{\kappa(1 \wedge t)} \mathrm{d}t \lesssim \kappa^{-1}(T + \log \delta^{-1}).$$

$\square$

## C.4 OVERALL ERROR BOUND

**Theorem C.6.** *Let $\overleftarrow{p}_{0:T}$ and $\widehat{q}_{0:T}$ be the path measures of the backward process with the stochastic integral formulation (3.2) and the interpolating process (A.15) of $\tau$-leaping algorithm (Algorithm (1)), then it holds that*

$$D_{\mathrm{KL}}(\overleftarrow{p}_{0:T} \| \widehat{q}_{0:T}) \le D_{\mathrm{KL}}(\overleftarrow{p}_0 \| \widehat{q}_0) + \mathbb{E}\left[\int_0^T \int_{\mathbb{X}} \left(\mu_s(y) \log \frac{\mu_s(y)}{\widehat{\mu}^\theta_{\lfloor s \rfloor}(y)} - \mu_s(y) + \widehat{\mu}^\theta_{\lfloor s \rfloor}(y)\right) \nu(\mathrm{d}y)\mathrm{d}t\right],$$
(C.2)

*where the expectation is taken w.r.t. paths generated by the backward process (3.2).*

*Proof.* The result directly follows by the arguments in the proof of Corollary 3.4 in Appendix A.3 and invoking the change of measure formula (Theorem 3.3) w.r.t. the true intensity $\mu_s$ as in (3.2) and the approximated intensity $\widehat{\mu}^\theta_{\lfloor s \rfloor}$ as in (4.2), Proposition 4.1. $\square$

Now, we are ready to present the proof of Theorem 4.7.

*Proof of Theorem 4.7.* We first rewrite the integral in (C.3) as

$$
\int_{s_n}^{s_{n+1}} \int_{\mathbb{X}} \left( \mu_s(y) \log \frac{\mu_s(y)}{\widehat{\mu}^\theta_{\lfloor s \rfloor}(y)} - \mu_s(y) + \widehat{\mu}^\theta_{\lfloor s \rfloor}(y) \right) \nu(\mathrm{d}y)\mathrm{d}s
$$

$$
= \int_{s_n}^{s_{n+1}} \int_{\mathbb{X}} \left( \mu_{s_n}(y) \log \frac{\mu_{s_n}(y)}{\widehat{\mu}^\theta_{\lfloor s_n \rfloor}(y)} - \mu_{s_n}(y) + \widehat{\mu}^\theta_{\lfloor s \rfloor}(y) \right.
$$

$$
\left. + G(\mu_s(y); \widehat{\mu}^\theta_{\lfloor s_n \rfloor}(y)) - G(\mu_{s_n}(y); \widehat{\mu}^\theta_{\lfloor s \rfloor}(y)) \right) \nu(\mathrm{d}y)\mathrm{d}s.
$$

Therefore, the overall error is bounded by

$$
D_{\mathrm{KL}}(\breve{p}_{0:T-\delta} \| \widehat{q}_{0:T-\delta})
$$

$$
\lesssim D_{\mathrm{KL}}(\breve{p}_0 \| \widehat{q}_0) + \mathbb{E}\left[ \int_0^{T-\delta} \int_{\mathbb{X}} \left( \mu_{s_n}(y) \log \frac{\mu_{s_n}(y)}{\widehat{\mu}^\theta_{\lfloor s_n \rfloor}(y)} - \mu_{s_n}(y) + \widehat{\mu}^\theta_{\lfloor s_n \rfloor}(y) \right) \nu(\mathrm{d}y)\mathrm{d}t \right]
$$

$$
+ \int_{s_n}^{s_{n+1}} \int_{\mathbb{X}} \left| G(\mu_s(y); \widehat{\mu}^\theta_{\lfloor s_n \rfloor}(y)) - G(\mu_{s_n}(y); \widehat{\mu}^\theta_{\lfloor s \rfloor}(y)) \right| \nu(\mathrm{d}y)\mathrm{d}s
$$

$$
\lesssim D_{\mathrm{KL}}(p_{T-\delta} \| p_\infty)
$$

$$
+ \mathbb{E}\left[ \sum_{n=0}^{N-1} (s_{n+1} - s_n) \right.
$$

$$
\left. \int_{\mathbb{X}} \left( \breve{s}_{s_n}(\breve{x}_{s_n^-}, y) \log \frac{\breve{s}_{s_n}(\breve{x}_{s_n^-}, y)}{\widehat{s}^\theta_{s_n}(\breve{x}_{s_n^-}, y)} - \breve{s}_{s_n}(\breve{x}_{s_n^-}, y) + \widehat{s}^\theta_{s_n}(\breve{x}_{s_n^-}, y) \right) \widetilde{Q}(\breve{x}_{s_n^-}, y)\nu(\mathrm{d}y) \right]
$$

$$
+ \sum_{n=0}^{N-1} (\log C + \log M) \overline{D}^2 \kappa (s_{n+1} - s_n)
$$

$$
\lesssim \begin{cases} \exp(-\rho T) \log |\mathbb{X}| + \epsilon + \overline{D}^2 \kappa T, & \gamma < 1, \\ \exp(-\rho T) \log |\mathbb{X}| + \epsilon + \overline{D}^2 \kappa \left( T + \log^2 \delta^{-1} \right), & \gamma = 1, \end{cases}
$$

where in the last inequality we used results for the first term (Truncation error, *cf.* Theorem C.1), the second term (Approximation error, *cf.* Assumption 4.6) and the third term (Discretization error, *cf.* Proposition C.5).

By taking

$$
T = \mathcal{O}\left( \frac{\log\left(\epsilon^{-1} \log |\mathbb{X}|\right)}{\rho} \right), \quad \kappa = \mathcal{O}\left( \frac{\epsilon\rho}{\overline{D}^2 \log\left(\epsilon^{-1} \log |\mathbb{X}|\right)} \right),
$$

deploying the time discretization scheme with $\gamma < \eta \lesssim 1 - T^{-1}$ when $\gamma < 1$, and $\eta = 1$ when $\gamma = 1$, and performing early stopping as

$$
\delta = \begin{cases} 0, & \gamma < 1, \\ \Omega\left( \exp(-\sqrt{T}) \right), & \gamma = 1, \end{cases}
$$

we have $D_{\mathrm{KL}}(\breve{p}_{T-\delta} \| \widehat{q}_T) \leq D_{\mathrm{KL}}(\breve{p}_{0:T-\delta} \| \widehat{q}_{0:T-\delta}) \lesssim \epsilon$ with

$$
N \lesssim \kappa^{-1} T = \mathcal{O}\left( \frac{\overline{D}^2 \log^2\left(\epsilon^{-1} \log |\mathbb{X}|\right)}{\epsilon\rho^2} \right)
$$

steps with probability

$$
1 - \mathcal{O}\left( \kappa \overline{D}^2 T \right) = 1 - \mathcal{O}(\epsilon).
$$

The proof is complete. □

**Theorem C.7.** *Let $\breve{p}_{0:T}$ and $\widehat{q}_{0:T}$ be the path measures of the backward process with the stochastic integral formulation (3.2) and the approximated backward process (4.4) of the uniformization algorithm (Algorithm (2)), then it holds that*

$$D_{\mathrm{KL}}(\breve{p}_{0:T}\|q_{0:T}) \leq D_{\mathrm{KL}}(\breve{p}_0\|q_0) + \mathbb{E}\left[\int_0^T \int_{\mathbb{X}} \left(\mu_s(y)\log\frac{\mu_s(y)}{\widehat{\mu}_s^\theta(y)} - \mu_s(y) + \widehat{\mu}_s^\theta(y)\right)\nu(\mathrm{d}y)\mathrm{d}t\right],$$
(C.3)

*where the expectation is taken w.r.t. paths generated by the backward process (3.2).*

*Proof.* The result directly follows by the arguments in the proof of Corollary 3.4 in Appendix A.3 and invoking the change of measure formula (Theorem 3.3) w.r.t. the true intensity $\mu_s$ as in (3.2) and the approximated intensity $\widehat{\mu}_s^\theta$ as in (4.4), Proposition 4.2. $\square$

*Proof of Theorem 4.9.* Due to the equivalence of the stochastic integral formulation (4.4) of the uniformization scheme and the approximate backward process (A.12) established in Proposition 4.2, the error for the uniformization scheme is directly bounded by the error (C.3) in Corollary 3.4, *i.e.*

$$D_{\mathrm{KL}}(\breve{p}_{0:T-\delta}\|q_{0:T-\delta})$$
$$\leq D_{\mathrm{KL}}(\breve{p}_0\|q_0) + \mathbb{E}\left[\int_0^{T-\delta}\int_{\mathbb{X}}\left(\mu_s(y)\log\frac{\mu_s(y)}{\widehat{\mu}_s^\theta(y)} - \mu_s(y) + \widehat{\mu}_s^\theta(y)\right)\nu(\mathrm{d}y)\mathrm{d}t\right]$$
$$\lesssim D_{\mathrm{KL}}(p_T\|p_\infty)$$
$$+ \mathbb{E}\left[\int_0^{T-\delta}\int_{\mathbb{X}}\left(\breve{s}_s(\breve{x}_{s^-},y)\log\frac{\breve{s}_s(\breve{x}_{s^-},y)}{\widehat{s}_s^\theta(\breve{x}_{s^-},y)} - \breve{s}_s(\breve{x}_{s^-},y) + \widehat{s}_s^\theta(\breve{x}_{s^-},y)\right)\widetilde{Q}(\breve{x}_{s^-},y)\nu(\mathrm{d}y)\mathrm{d}s\right]$$
$$\lesssim |\mathbb{X}|\exp(-\rho T) + \epsilon.$$

The expectation of the number of steps $N$ is bounded by

$$\mathbb{E}[N] = \mathbb{E}\left[\sum_{b=0}^{B-1}\mathcal{P}\left(\overline{\lambda}_{s_{b+1}}(s_{b+1} - s_b)\right)\right] = \sum_{b=0}^{B-1}\overline{\lambda}_{s_{b+1}}(s_{b+1} - s_b)$$
$$\lesssim \sum_{b=0}^{B-1}\overline{D}(1\vee(T - s_{b+1}))^{-1}(s_{b+1} - s_b)$$
$$\lesssim \overline{D}\left(T + \int_\delta^1 t^{-1}\mathrm{d}t\right) = \overline{D}(T + \log\delta^{-1}).$$

By taking

$$T = \mathcal{O}\left(\frac{\log\left(\epsilon^{-1}\log|\mathbb{X}|\right)}{\rho}\right), \quad \delta = \Omega\left(\exp(-T)\right)$$

we have $D_{\mathrm{KL}}(\breve{p}_{T-\delta}\|q_{T-\delta}) \leq D_{\mathrm{KL}}(\breve{p}_{0:T-\delta}\|q_{0:T-\delta}) \lesssim \epsilon$ with

$$\mathbb{E}[N] = \mathcal{O}\left(\frac{\overline{D}\log\left(\epsilon^{-1}\log|\mathbb{X}|\right)}{\rho}\right)$$

steps. $\square$

