# OpenReview forum: "How Discrete and Continuous Diffusion Meet: Comprehensive Analysis of Discrete Diffusion Models  via a Stochastic Integral Framework"
_ICLR.cc/2025/Conference — ICLR 2025 Poster_

### Official Review · Reviewer_5uy7 · 2024-10-30

**Soundness:** 3
**Presentation:** 3
**Contribution:** 3
**Rating:** 8
**Confidence:** 2

**Summary:**

This paper investigates the error analysis of discrete diffusion models. To this end, the authors develop a framework based on Levy-type stochastic integrals and establish a stochastic integral formulation of discrete diffusion models. The framework is utilized to derive the first error bound and sheds insight into the design of efficient and accurate discrete diffusion models.

**Strengths:**

1. This paper proposes the first error bound for the diffusion process, which provides insights and guidance for future research on discrete diffusion models.

2. Well-designed examples are presented for better illustration.

3. Rigorous theoretical analyses establish the foundation of the framework and the error bound for discrete diffusion models.

**Weaknesses:**

1. This paper is of theoretical interest. Some simulation experimental results are suggested to confirm the error analysis if possible.

2. This paper is mathematic-heavy. Some insights, intuitions, or illustrations are suggested to provide for better comprehension.

3. The error analysis in Theorem 4.7 is built on four assumptions, which seems to weaken the practicality of the error bound. I suggest the authors justify or explain this point.

**Questions:**

Please refer to the weakness part.

---

> ### Author Response · Authors · 2024-11-19
>
> We thank the reviewer for their insightful comments and constructive feedback. Below, we address each point raised in a detailed manner.
>
> ---
>
> ## Regarding Experimental Results
>
> Please refer to the common response ("Regarding Empirical Validation") for our approach to incorporating empirical validation of the error bounds. We plan to conduct numerical experiments that will further substantiate our theoretical findings and illustrate their practical relevance.
>
> ---
>
> ## Regarding the Presentation
>
> Please refer to the common response ("Regarding the Presentation") for our comprehensive plan to improve the paper's readability and accessibility. We aim to enhance the clarity of our presentation by including more intuitive explanations and examples, making the content more approachable for readers with varying levels of familiarity with the subject matter.
>
> ---
>
> ## Regarding the Assumptions in Theorem 4.7
>
> We appreciate the reviewer's detailed inquiry into the assumptions underpinning Theorem 4.7. These assumptions are crucial for ensuring the robustness and applicability of our theoretical results, and we offer the following elaborations to clarify their necessity and scope:
> - **Assumption 4.3**: (i) This assumption pertains to the well-definedness and regularity of the rate matrix $\boldsymbol Q$, which is a standard requirement and typically satisfied in scenarios like fully connected graph structures [1] or models with uniform rates [2]. (ii) It translates to the connectivity of the graph $\mathcal G(\boldsymbol Q)$, analogous to the assumptions underlying the exponential convergence of the OU process in continuous models. While often implicitly assumed, our explicit formulation provides a method to verify and quantify this convergence.
> - **Assumption 4.4**: As justified in Remark B.3, the assumption is mainly presuming that the value of the score function $\boldsymbol{s}_t$ is at most of order $\mathcal O(t{-1})$, which is also assumed in previous work [3]. In many cases, such assumption can be relaxed to $\mathcal O(1)$ (*cf.* Assumption 2 [2]), *e.g.* by assuming the target data distribution to be both lower and upper bounded (*cf.* Assumption 3 [3] and Assumption 2.4 [4] for the case of continuous diffusion models). The bound on the neural network-based score function is to ensure well-posed behavior of the approximate reverse process, which can either be strictly enforced via manual truncation or implicitly imposed by regularization techniques during training. Similar assumptions are also made on the neural network approximations for the case of continuous diffusion models (*cf.* Assumption 3 [5], Assumption 3.3 [6] and Assumption 4.3 [7]).
> - **Assumption 4.5**: Suppose $Q(x_{t-}, y) = \Theta(1)$, this assumption is trivially satisfied for $\gamma = 1$ given Assumption 4.3. However, we choose to introduce the parameter $\gamma\in[0, 1]$, which represents the local continuity of the score function (*cf.* the assumption on Lipschitz continuity of the true score function for continuous diffusion models, *e.g.* Assumption 2 [5], Assumption 4.2 [7], Assumption 1 [8] and Assumption 3 [9]), to investigate how the error bound scales with this continuity measure. As shown in Theorem 4.7, we can see that one may need different time discretization schemes for different levels of continuity.
> - **Assumption 4.6**: This assumption is essentially stating that the neural network based score estimator is $\epsilon$-accurate in terms of the score entropy loss, which also appeared as Assumption 1 in [3]. In fact, this assumption can be treated as an analog of the $L^2$-accuracy assumption, which is widely adopted in related work on the theoretical analysis of continuous diffusion models (*cf.* Assumption 4 [5], Assumption 3.2 [6], Assumption 3 [8] and Assumption 1 [9]).
>
> These assumptions ensure the theoretical soundness of our framework and are aligned with practical conditions often encountered in the implementation of diffusion models. We believe these conditions are reasonable and pragmatic, though future work may explore their relaxation to enhance the framework's flexibility and applicability.

---

> ### Author Response · Authors · 2024-11-19
>
> We hope this response satisfactorily addresses the reviewer’s concerns and clarifies the depth and rigor of our approach. We are open to further discussions and are grateful for the opportunity to enhance our manuscript based on your feedback.
>
> ---
>
> ### References
>
> [1] Lou, Aaron, Chenlin Meng, and Stefano Ermon. "Discrete Diffusion Modeling by Estimating the Ratios of the Data Distribution." Forty-first International Conference on Machine Learning.\
> [2] Campbell, Andrew, et al. "A continuous time framework for discrete denoising models." Advances in Neural Information Processing Systems 35 (2022): 28266-28279.\
> [3] Chen, Hongrui, and Lexing Ying. "Convergence analysis of discrete diffusion model: Exact implementation through uniformization." arXiv preprint arXiv:2402.08095 (2024).\
> [4] Oko, Kazusato, Shunta Akiyama, and Taiji Suzuki. "Diffusion models are minimax optimal distribution estimators." In International Conference on Machine Learning, pp.26517-26582. PMLR, 2024\
> [5] Chen, Sitan, et al. "The probability flow ode is provably fast." Advances in Neural Information Processing Systems 36 (2024).\
> [6] Huang, Daniel Zhengyu, Jiaoyang Huang, and Zhengjiang Lin. "Convergence analysis of probability flow ODE for score-based generative models." arXiv preprint arXiv:2404.09730 (2024).\
> [7] Dou, Zehao, et al. "Theory of Consistency Diffusion Models: Distribution Estimation Meets Fast Sampling." In International Conference on Machine Learning, pp.11592-11612. PMLR, 2024.\
> [8] Chen, Sitan, et al. "Sampling is as easy as learning the score: theory for diffusion models with minimal data assumptions." In: The Eleventh International Conference on Learning Representations. 2023.\
> [9] Chen, Hongrui, Holden Lee, and Jianfeng Lu. "Improved analysis of score-based generative modeling: User-friendly bounds under minimal smoothness assumptions." International Conference on Machine Learning. PMLR, 2023.

---

> > ### Comment · Reviewer_5uy7 · 2024-11-26
> >
> > Thanks for the detailed responses. My concerns are addressed.

---

### Official Review · Reviewer_5nb2 · 2024-11-03

**Soundness:** 3
**Presentation:** 2
**Contribution:** 3
**Rating:** 6
**Confidence:** 2

**Summary:**

The paper provides a framework to analyze the KL-error of discrete diffusion models using a stochastic integral approach. The authors introduce a formulation for discrete diffusion processes using Poisson random measures with time- and state-dependent intensities. This framework allows them to decompose the error into truncation, approximation, and discretization errors. The key contributions include considering the $\tau$-leaping scheme and establishing an error bound for it in terms of KL divergence.

**Strengths:**

The integral formulation for discrete diffusion models is insightful and provides strong motivation for the proposed algorithms. The error bounds in theorems 4.7 and 4.9 are neat.

**Weaknesses:**

While this is a theory-focused paper, some experimental evaluation would enhance the work, particularly to demonstrate the practical accuracy of the error bounds or the performance of Algorithm 1 and Algorithm 2 in generative applications.

The presentation is highly technical from the outset, assuming a strong theoretical background in diffusion models. The extensive use of dense notation and specialized terminology makes the paper challenging to read and less accessible.

**Questions:**

How important is the request "Q symmetric" for assumptions 4.3--4.6?

---

> ### Author Response · Authors · 2024-11-19
>
> We thank the reviewer for their constructive feedback and for highlighting both the strengths and areas for improvement in our work. We address the concerns raised below.
>
> ---
>
> ## Regarding the Experimental Evaluation
>
> Please refer to the common response ("Regarding Empirical Validation") for our detailed response to the request for numerical experiments. In brief, while this work focuses on theoretical contributions, we acknowledge the value of empirical validation and plan to explore numerical experiments to assess the practical accuracy of our error bounds and algorithmic performance in future work.
>
> ---
>
> ## Regarding the Presentation
>
> Please refer to the common response ("Regarding the Presentation") for our detailed plan to improve the readability and accessibility of the paper. In summary, we will enhance the explanations, include illustrative examples, and reorganize sections to make the content more intuitive and approachable for a broader audience.
>
> ---
>
> ## Regarding the Assumption of Symmetry for $\boldsymbol Q$
>
> We appreciate the reviewer’s insightful question about the role of the symmetry assumption for $\boldsymbol Q$ in our analysis. Below, we provide a detailed clarification:
>
> - **Applicability in Practice**: The assumption of $\boldsymbol Q$ being symmetric is consistent with a wide range of discrete diffusion model designs commonly used in practice, such as models with fully connected graph structures [1] or uniform rates [2].
> - **Assumption-Specific Dependencies**:
>     - Assumptions 4.3(i), 4.5, and 4.6 are independent of the symmetry of $\boldsymbol Q$.
>     - Assumption 4.3(ii), concerning the lower bound on the modified log-Sobolev constant $\rho(\boldsymbol Q)$, is generally related to the connectivity and structural properties of the graph $\mathcal G(\boldsymbol Q)$ but not directly related to symmetry.
>     - Assumption 4.4 leverages symmetry in our proof (see Remark B.3), as it simplifies certain derivations. However, these arguments can be extended to non-symmetric $\boldsymbol Q$ by introducing additional spectral assumptions on $\boldsymbol Q$. We will discuss this extension in the revised version of the paper.
> - **General Applicability of Our Framework**: Our core theoretical contributions, including the stochastic integral formulation (Propositions 3.2, 4.1, 4.2) and the change of measure arguments (Theorem 3.3, Corollary 3.5), are **independent** of the symmetry assumption for $\boldsymbol Q$. This ensures that the foundation of our error analysis holds for non-symmetric $\boldsymbol Q$ as well. The assumption of $\boldsymbol Q$ being symmetric only simplifies the proof of the discretization error (Proposition C.5).
>
> We strongly believe that our error bounds can be extended to non-symmetric $\boldsymbol Q$ under appropriate assumptions, including cases involving absorbing states [1, 3]. This represents an interesting direction for future work, and we are excited to explore these generalizations further.
>
> ---
>
> We once again thank the reviewer for their thoughtful comments, which have helped us refine and clarify our work. We hope our responses address the concerns raised, and we are happy to provide further clarifications or explanations as needed.
>
> ---
> ### References
>
> [1] Lou, Aaron, Chenlin Meng, and Stefano Ermon. "Discrete Diffusion Modeling by Estimating the Ratios of the Data Distribution." Forty-first International Conference on Machine Learning.\
> [2] Campbell, Andrew, et al. "A continuous time framework for discrete denoising models." Advances in Neural Information Processing Systems 35 (2022): 28266-28279.\
> [3] Ou, Jingyang, et al. "Your Absorbing Discrete Diffusion Secretly Models the Conditional Distributions of Clean Data." arXiv preprint arXiv:2406.03736 (2024).

---

### Official Review · Reviewer_Se7m · 2024-11-04

**Soundness:** 4
**Presentation:** 3
**Contribution:** 3
**Rating:** 8
**Confidence:** 3

**Summary:**

The paper develops a rigorous theoretical foundation for discrete diffusion models, drawing parallels with continuous diffusion models. The authors introduce a novel stochastic integral framework using Lévy-type integrals, which enables a structured representation of discrete diffusion processes. They establish change-of-measure techniques analogous to Girsanov’s theorem, facilitating error analysis in terms of KL divergence.
By decomposing errors into truncation, approximation, and discretization components, the paper provides the first KL divergence bounds for the τ-leaping scheme in discrete settings. This unified framework also compares $\tau$-leaping and uniformization methods, highlighting computational efficiency and accuracy. Overall, the work advances the theoretical understanding of discrete diffusion models, making it possible to design more accurate and efficient algorithms for real-world applications that require discrete data modeling.

**Strengths:**

1.Theorem 3.3 establishes a change of measure theorem for Poisson random measures with evolving intensity, the authors introduce a discrete analog to Girsanov's theorem. This is a breakthrough, as it enables KL divergence analysis for discrete models, a theoretical advancement that makes error analysis feasible for the discrete setting.
2.Section 4, the paper follows a classical error analysis for diffusion models, breaking down error into truncation, approximation, and discretization components. This analysis, grounded in Theorems 4.7 and 4.9, is particularly valuable as it allows for practical application through the $\tau$-leaping and uniformization algorithms, with explicit convergence guarantees.
3.The paper presents the first KL divergence bounds for the t-leaping scheme in discrete diffusion models. This error bound is stronger and more informative than prior work using total variation distance.

**Weaknesses:**

I don't see major weaknesses of this work, but I have some comments:
1. The paper provides rigorous theoretical error bounds for τ-leaping and uniformization (Theorems 4.7 and 4.9). Do the authors plan to empirically validate these bounds on synthetic or real-world datasets?
2. Could the authors provide runtime or memory complexity comparisons for τ-leaping versus uniformization under varying parameters?
3. The paper centers on Lévy-type integrals for discrete diffusion models. Could the authors comment on other potential stochastic frameworks for discrete models and when the proposed Lévy-type integral framework would be preferable over alternatives?

**Questions:**

see above Weaknesses section.

---

> ### Author Response · Authors · 2024-11-19
>
> We sincerely thank the reviewer for their positive feedback and insightful questions, which have provided valuable opportunities to clarify and expand upon key aspects of our work. We address the comments point by point below.
>
> ---
>
> ## Regarding Empirical Validation of Error Bounds
>
> We appreciate the reviewers’ concerns about empirical validation and refer to the common response ("Regarding Empirical Validation") for detailed plans to incorporate numerical experiments that validate our theoretical error bounds and compare algorithmic performance in future work.
>
> ---
>
> ## Regarding Runtime and Memory Complexity Comparisons
>
> We appreciate the reviewer's interest in runtime and memory complexity comparisons. In the current version of the paper, we provide a runtime comparison following Theorem 4.9, highlighting a potential advantage of the uniformization algorithm in reducing computational cost. This analysis underscores how our theoretical framework can guide practitioners in selecting more efficient algorithms for simulating the backward process.
>
> As for memory complexity, if the reviewer is referring to the memory required to store the neural network used for approximating the score function, the memory complexities of the $\tau$-leaping and uniformization algorithms are identical. If a different aspect of memory complexity is intended, we would be glad to investigate and address it in further detail.
>
> ---
>
> ## Regarding Other Potential Stochastic Frameworks
>
> We thank the reviewer for their astute observation regarding alternative stochastic frameworks for discrete diffusion models. For Markov processes, two main formulations exist: the distribution-based and the path-based (e.g., forward and backward processes represented by state distributions or by state trajectories over time). In continuous diffusion models, the path-based formulation using stochastic differential equations (SDEs [3]) is favored for its intuitive interpretation and utility in both implementation and theoretical analysis.
>
> In contrast, current analyses of discrete diffusion models predominantly use the continuous-time Markov chain (CTMC) framework [1, 2]. While effective, this approach is less intuitive for theoretical analysis and lacks a clear connection to continuous diffusion models. Our work introduces a path-based formulation for discrete diffusion models through Lévy-type stochastic integrals, which parallels the SDE framework in continuous diffusion. This formulation not only bridges the theoretical gap between discrete and continuous diffusion models but also facilitates unified error analysis for algorithms like $\tau$-leaping and uniformization.
>
> Moreover, Lévy processes, characterized by infinite divisibility and the Lévy-Khintchine theorem, encompass a drift, Brownian motion, and jump process, allowing for broad applicability [4]. In discrete state spaces, where drifts and Brownian motions are not applicable, Lévy-type integrals simplify to stochastic integrals with respect to Poisson random measures. This makes our framework both general and well-suited for discrete diffusion models. Potential alternatives could include removing Markov assumptions, a direction we believe holds promise for future research. Additionally, exploring how diffusion models based on different Markov processes perform across various tasks would be an intriguing area of practical investigation [5, 6, 7].
>
> ---
>
> In conclusion, we once again thank the reviewer for their thoughtful feedback and questions, which have greatly helped in clarifying the scope and implications of our work. We hope our responses have addressed all concerns and are happy to provide further clarifications or elaborations if needed.
>
> ---
> ### References
>
> [1] Campbell, Andrew, et al. "A continuous time framework for discrete denoising models." Advances in Neural Information Processing Systems 35 (2022): 28266-28279.\
> [2] Chen, Hongrui, and Lexing Ying. "Convergence analysis of discrete diffusion model: Exact implementation through uniformization." arXiv preprint arXiv:2402.08095 (2024).\
> [3] Song, Yang, et al. "Score-based generative modeling through stochastic differential equations." arXiv preprint arXiv:2011.13456 (2020).\
> [4] Benton, Joe, et al. "From denoising diffusions to denoising Markov models." Journal of the Royal Statistical Society Series B: Statistical Methodology 86.2 (2024): 286-301.\
> [5] Yoon, Eun Bi, et al. "Score-based generative models with Lévy processes." Advances in Neural Information Processing Systems 36 (2023): 40694-40707.\
> [6] Chen, Yifan, et al. "Probabilistic Forecasting with Stochastic Interpolants and F\" ollmer Processes." In International Conference on Machine Learning, pp.6728-6756. PMLR, 2024.\
> [7] Winkler, Ludwig, Lorenz Richter, and Manfred Opper. "Bridging discrete and continuous state spaces: Exploring the Ehrenfest process in time-continuous diffusion models." In International Conference on Machine Learning, pp.53017-53038. PMLR, 2024.

---

### Official Review · Reviewer_XLAW · 2024-11-07

**Soundness:** 3
**Presentation:** 2
**Contribution:** 2
**Rating:** 6
**Confidence:** 2

**Summary:**

This paper addresses the challenge of error analysis in discrete diffusion
models. To bridge the gap between discrete and continuous diffusion models
(for which the error analysis is better understood), the authors propose a
framework based on Levy-type stochastic integrals, by generalizing Poisson
random measures to support state-dependent and time-varying intensities
under assumptions of regularity of rate matrix, and bounded and continuous
score function. This framework allows for error decomposition into
components such as truncation, approximation, and discretization errors,
providing clearer insight into error sources.  The analysis is performed
for tau-leaping and uniformization methods, two methods to simulate the
backward process. They establish stronger error bounds compared to previous
work using KL divergence.

**Strengths:**

Pros:
- Very non-trivial theoretical work.

**Weaknesses:**

Cons:
- Positioning of the work is not clear. It is not compared clearly with
  previous work.
- Motivation and implications for practical usage not provided.
- Very dense writing. Hard to understand.

Details:
The main results are two theorems, one for tau-leaping and the other for
uniformization. No attempt is made to provide outlines of the proofs. What
is the basic motivation for this research? How does it contribute to the
field (reduce algorithmic complexity, improve guarantees, etc.)?
The reader could greatly benefit from proof outlines.

The paper is very hard to understand. For a reader who is not deeply
immersed in this topic, it is almost impossible to understand the
implications, general approaches, and the proofs themselves.

The third bullet in contributions says that the work unifies and fortifies
existing research on discrete diffusion models. The reviewer did not come
across any statement in support of this or elaborating this point.

Related works does not cover any theoretical work on discrete diffusion
models. The introduction does mention several papers in this regard.
Organization can be made better.

The paper starting at Preliminaries is very dense. That said, the authors
have put effort to state the assumptions clearly and provided discussions
on differences between continuous and discrete diffusion models w.r.t. the
various errors considered.

**Questions:**

Questions inserted in weaknesses.

---

> ### Author Response · Authors · 2024-11-19
>
> We thank the reviewer for their thoughtful and constructive feedback, which has provided valuable insights into improving the clarity, positioning, and contributions of our work. We address the points raised below in detail.
>
> ---
>
> ## Regarding the Positioning and Motivation of the Work
>
> We appreciate the reviewer's concerns about the positioning and motivation of our work. As highlighted in the introduction, the primary motivation stems from the lack of a systematic and rigorous theoretical framework for discrete diffusion models, compared to the well-established literature on continuous diffusion models. Our contributions build on the following works:
> - **Methodology**: Construction of forward and backward processes in discrete state spaces [1, 2], ratio matching [3], score entropy loss [4, 6], denoising score entropy loss [3].
> - **Error Analysis for Discrete Diffusion**: CTMC-based error analysis for $\tau$-leaping in TV distance [4], Feller process-based analysis for general denoising Markov models [6], and recent work on uniformization algorithms for $\mathbb{X} = \\{ 0,1 \\}^d$ [5].
> - **Error Analysis for Continuous Diffusion**: Sampling guarantees in KL divergence [7, 8, 9] and TV distance [10, 11].
>
> By consolidating and extending these works, we aim to establish a comprehensive theoretical framework for discrete diffusion models, addressing the fragmented nature of the current literature.
>
> ---
>
> ## Regarding the Related Works
>
> We recognize the reviewer's point about the organization of related works. Given the scarcity of theoretical works on discrete diffusion models, we focused our discussion on two key references [4, 5] within the introduction to motivate and contextualize our contributions. However, we acknowledge that this organization may have diluted the clarity of positioning. In the revised manuscript, we will reorganize the introduction and related works sections to better articulate our contributions and how they relate to prior literature.
>
> ---
>
> ## Regarding the Contributions
>
> We acknowledge the reviewer's concerns regarding the clarity of the contributions, especially as claimed in the third bullet point of Section 1.1 (Contributions). We agree with the reviewer that the contributions should be more clearly and explicitly articulated. Below, we elaborate on the novel contributions of our work:
>
> - **Technique Advancement**: We introduce Poisson random measures with evolving intensities, formulating discrete diffusion models as stochastic integrals. This approach, supported by a novel Girsanov's theorem, provides a fresh perspective on analyzing discrete diffusion models.
> - **Unified and Fortified Error Analysis**: Our methodology is capable of deriving error bounds and thus unifying the error analysis for both the $\tau$-leaping [4] and uniformization algorithms [5], under relaxed assumptions. Importantly, we establish the first theoretical guarantees for the $\tau$-leaping algorithm in KL divergence, a significant improvement over prior work in TV distance.
> - **Bridging Discrete and Continuous Models**: Our framework connects the error analysis of discrete and continuous diffusion models, facilitating the transfer of theoretical and practical insights between these domains. This thus sheds light on establishing convergence guarantees for a broader range of discrete diffusion models.
>
> We will rewrite the contributions section in the revised manuscript to incorporate these points more explicitly, addressing the feedback from the reviewer.
>
> ---
>
> ## Regarding the Implications for Practical Usage
>
> We thank the reviewer for pointing out the need to better highlight the practical implications of our work. While our paper is primarily theoretical, we analyze the $\tau$-leaping and uniformization algorithms, two widely used methods for inference in discrete diffusion models. Our error analysis provides key insights into the convergence properties and computational complexity of these algorithms. To clarify these implications, we will:
> - Expand the discussion following Theorem 4.9 to explicitly compare the runtime of the $\tau$-leaping and uniformization algorithms.
> - Emphasize how our theoretical framework can guide the design and analysis of new algorithms for training and inference in discrete diffusion models.
>
> In summary, we believe our work provides a unified framework to analyze and compare the time complexity of algorithms used for simulating the backward process, with potential to inform future advancements in the field.

---

> ### Author Response · Authors · 2024-11-19
>
> ## Regarding the Presentation
>
> We acknowledge the concerns about the paper's dense writing and technical detail. Please refer to our common response ("Regarding the Presentation") for specific plans to enhance the paper's accessibility, including the addition of intuitive explanations, examples, and proof outlines to aid understanding.
>
> ---
>
> In conclusion, we sincerely thank the reviewer for their detailed comments and questions, which have greatly helped us identify areas for improvement. We will incorporate these suggestions into the revised version of the paper. Should there be further questions or concerns, we would be happy to address them.
>
> ---
> ### References
> [1] Austin, Jacob, et al. "Structured denoising diffusion models in discrete state-spaces." Advances in Neural Information Processing Systems 34 (2021): 17981-17993.\
> [2] Sun, Haoran, et al. "Score-based continuous-time discrete diffusion models." arXiv preprint arXiv:2211.16750 (2022).\
> [3] Lou, Aaron, Chenlin Meng, and Stefano Ermon. "Discrete Diffusion Modeling by Estimating the Ratios of the Data Distribution." Forty-first International Conference on Machine Learning.\
> [4] Campbell, Andrew, et al. "A continuous time framework for discrete denoising models." Advances in Neural Information Processing Systems 35 (2022): 28266-28279.\
> [5] Chen, Hongrui, and Lexing Ying. "Convergence analysis of discrete diffusion model: Exact implementation through uniformization." arXiv preprint arXiv:2402.08095 (2024).\
> [6] Benton, Joe, et al. "From denoising diffusions to denoising markov models." arXiv preprint arXiv:2211.03595 (2022).\
> [7] Chen, Sitan, et al. "Sampling is as easy as learning the score: theory for diffusion models with minimal data assumptions." In: The Eleventh International Conference on Learning Representations. 2023.\
> [8] Chen, Hongrui, Holden Lee, and Jianfeng Lu. "Improved analysis of score-based generative modeling: User-friendly bounds under minimal smoothness assumptions." International Conference on Machine Learning. PMLR, 2023.\
> [9] Benton, Joe, et al. "Linear convergence bounds for diffusion models via stochastic localization."  In: The Eleventh International Conference on Learning Representations. 2024.\
> [10] Chen, Sitan, et al. "The probability flow ode is provably fast." Advances in Neural Information Processing Systems 36 (2024).\
> [11] Huang, Daniel Zhengyu, Jiaoyang Huang, and Zhengjiang Lin. "Convergence analysis of probability flow ODE for score-based generative models." arXiv preprint arXiv:2404.09730 (2024).

---

### Author Response · Authors · 2024-11-19
**Common Responses**

We sincerely thank all the reviewers for their constructive feedback and for recognizing the contributions of our work. We are encouraged by the positive comments describing our framework as "*insightful*" and a "*breakthrough*," with "*well-designed examples*" and "*stronger and more informative*" results. Below, we address two common concerns raised by multiple reviewers.

---

## Regarding the Presentation (Reviewers XLAW, 5nb2, 5uy7)

We acknowledge the reviewers' concerns about the paper's readability and accessibility. As a theoretical work, our primary objective is to develop a rigorous stochastic integral-based framework for discrete diffusion models. At the same time, we aim to draw clear comparisons with continuous diffusion models to provide insight into the implications and guarantees of our results. This dual focus inevitably involves significant mathematical depth and detailed notations.

However, we recognize that the presentation may have become overly dense, potentially making it less accessible to a broader audience who may not have the opportunity to delve into all the technical details. To address these issues, we propose the following revisions:

- **Section 1 (Introduction)**: We will clarify the differences between continuous and discrete diffusion models, emphasize the motivations for discrete formulations, and outline the challenges in their theoretical analysis compared to continuous counterparts.
- **Section 3 (Stochastic Integral Formulation)**: We will supplement the existing formal definitions with verbal explanations and illustrative examples. Some technical details will be moved to the appendix to improve the flow of the main text. To aid understanding, we will introduce examples demonstrating the properties of Poisson random measures with evolving intensities. We will also expand the discussion on the motivation for using Lévy-type integrals to provide a more intuitive interpretation.
- **Section 3.2 (Change of Measure)**: This section will be expanded to elaborate on the connection between the continuous and discrete frameworks. Specifically, we will clarify how the change of measure theorem parallels Girsanov's theorem for continuous models and its role in enabling the error analysis for the $\tau$-leaping and uniformization algorithms.
- **Section 4 (Error Analysis)**: We will provide clearer algorithmic descriptions and brief justifications for the $\tau$-leaping and uniformization methods. Additionally, we will outline the analytical challenges each algorithm presents, including their error decomposition into truncation, approximation, and discretization components.
- **Proof Sketches for Theorems**: Proof sketches for Theorems 4.7 and 4.9 will be included in the appendix to outline the key ideas and techniques used in the proofs, addressing the feedback about providing insights into the theoretical foundations of our results.

We believe these revisions will significantly enhance the accessibility of the paper without compromising its rigor. They will help readers better grasp the theoretical contributions and implications, and facilitate further research in discrete diffusion models.

---

> ### Author Response · Authors · 2024-11-19
>
> ## Regarding Empirical Validation (Reviewers Se7m, 5nb2, 5uy7)
>
> We thank the reviewers for emphasizing the importance of numerical experiments to complement our theoretical findings. While this paper primarily focuses on developing a mathematical framework and error analysis for discrete diffusion models, we agree that empirical validation would provide valuable insights into the practical implications of our results. We acknowledge that empirical studies, such as those by Austin et al. [1], Lou et al. [2], and Campbell et al. [3], have significantly advanced the practical aspects of diffusion models. In contrast, our work focuses on providing theoretical foundations that could inform and guide empirical research.
>
> Adding numerical experiments to compare the $\tau$-leaping and uniformization algorithms in simulating the backward continuous-time Markov chain with a fixed (pretrained) score function would be a logical next step. If time permits, we plan to include such numerical experiments in the revised version of the paper to strengthen the connection between our theoretical results and practical performance. In future research, we also aim to study the broader implications of our framework on algorithm design and analysis, potentially addressing practical concerns such as computational efficiency and algorithmic robustness.
>
> ---
> ### References
>
> [1] Austin, Jacob, et al. "Structured denoising diffusion models in discrete state-spaces." Advances in Neural Information Processing Systems 34 (2021): 17981-17993.\
> [2] Lou, Aaron, Chenlin Meng, and Stefano Ermon. "Discrete Diffusion Modeling by Estimating the Ratios of the Data Distribution." Forty-first International Conference on Machine Learning.\
> [3] Campbell, Andrew, et al. "A continuous time framework for discrete denoising models." Advances in Neural Information Processing Systems 35 (2022): 28266-28279.

---

### Author Response · Authors · 2024-11-25
**Revision Summary**

We appreciate all the reviewers for their constructive feedback, which have greatly helped improve the quality of our manuscript. We have revised our manuscript to enhance readability and accessibility as suggested and marked all modifications in blue. Notable changes include clearer motivations in the introduction, and enriched explanations and insights in Section 3. We have also streamlined the main text by relocating some complex details to the appendix, and adding proof sketches for key theorems in Appendix C.1 to clarify our theoretical approaches. These revisions aim to make our content more accessible and understandable without compromising its rigor. We are grateful for your insights and hope our revision has addressed their concerns on the presentation of our work.

---

### Meta-Review · Area_Chair_f5DX · 2024-12-21

**Metareview:**

In order to unify the theoretical analysis of continuous and discrete diffusion models, this paper introduces a stochastic integral formulation based on Lévy-type stochastic integrals and generalizes the Poisson random measure to one with time-independent and state-dependent intensity. It provides change of measure theorems analogous to Itô integrals and Girsanov's theorem for continuous diffusion models. The proposed framework unifies and strengthens existing theoretical results, offering the first error bound for the τ-leaping scheme in KL divergence. This paper has received unanimous support from the reviewers. Therefore, I recommend acceptance. In the camera-ready version, the authors are encouraged to include a more detailed comparison with the concurrent work by Zhang et al. (2024), as both works analyze K states for each discrete random variable/token. While the sampling algorithms differ (uniformization vs. τ-leaping), the main theorems appear similar, thus a comparison between them would be helpful.

**Additional Comments On Reviewer Discussion:**

This is a purely theoretical paper. The theoretical result appears strong because it provides the first error bound for the τ-leaping scheme in KL divergence. I believe there is already a consensus, even prior to the rebuttal. The practical value of this paper is limited, as it does not propose any new algorithms. Therefore, I recommend accepting it as a poster.

---

### Decision · Program_Chairs · 2025-01-22

Accept (Poster)